# Coevolutionary Continuous Discrete Diffusion:
# Make Your Diffusion Language Model a Latent Reasoner

**Cai Zhou** [1 2]   **Chenxiao Yang** [3]   **Yi Hu** [4]   **Chenyu Wang** [1]   **Chubin Zhang** [5]
**Muhan Zhang** [4]   **Lester Mackey** [2]   **Tommi Jaakkola** [1]   **Stephen Bates** [1]   **Dinghuai Zhang** [2]

## Abstract

Diffusion language models, especially masked discrete diffusion models, have achieved great success recently. While there are some theoretical and primary empirical results showing the advantages of latent reasoning with looped transformers or continuous chain-of-thoughts, continuous diffusion models typically underperform their discrete counterparts. In this paper, we argue that diffusion language models do not necessarily need to be in the discrete space. In particular, we prove that continuous diffusion models have stronger expressivity than discrete diffusions and looped transformers. We attribute the contradiction between the theoretical expressiveness and empirical performance to their practical trainability: while continuous diffusion provides intermediate supervision that looped transformers lack, they introduce additional difficulty decoding tokens into the discrete token space from the continuous representation space. We therefore propose **C**oevolutionary **C**ontinuous **D**iscrete **D**iffusion (CCDD), which defines a joint multimodal diffusion process on the union of a continuous representation space and a discrete token space, leveraging a single model to simultaneously denoise in the joint space. By combining two modalities, CCDD is expressive with rich semantics in the latent space, as well as good trainability and sample quality with the help of explicit discrete tokens. We also propose effective architectures and advanced training/sampling techniques for CCDD, which reveals strong empirical performance in extensive language modeling experiments on real-world tasks.

[1]Massachusetts Institute of Technology  [2]Microsoft Research  [3]Toyota Technological Institute at Chicago  [4]Peking University  [5]Tsinghua University. Correspondence to: Cai Zhou <caiz428@mit.edu>, Dinghuai Zhang <dinzhang@microsoft.com>.

*Proceedings of the 43$^{rd}$ International Conference on Machine Learning*, Seoul, South Korea. PMLR 306, 2026. Copyright 2026 by the author(s).

## 1. Introduction

Recent years have seen great success of autoregressive (AR) large language models (LLMs) (Achiam et al., 2023; Yang et al., 2024b; Liu et al., 2024), especially their significant improvement in complex reasoning (Guo et al., 2025; Comanici et al., 2025). However, the **discrete** and **left-to-right** nature of these models still poses some fundamental difficulties. It is a known result in computation complexity theory that the expressivity of transformers – the architectural foundations of modern LLMs, are restricted in the class $\mathsf{TC}^0$ even with logarithmic Chain-of-Thought (CoT) steps (Merrill & Sabharwal, 2025). This suggests that transformers cannot accurately address problems outside the $\mathsf{TC}^0$ class such as *recognizing formal language* which measures state tracking capabilities, and *graph connectivity* that captures multistep reasoning ability, under standard complexity conjectures. Empirically, even state-of-the-art LLMs often struggle in a wide range of complex tasks requiring strong planning, parallel searching, and backtracking capabilities, such as *Sudoku*. To overcome these challenges, researchers have been working to develop new language modeling paradigms.

On the one hand, LLMs are shown to be benefit from latent reasoning through various ways, including looped transformers (LT) (Giannou et al., 2023) or continuous CoT (Hao et al., 2024). One line of research on *looped transformers* or *universal transformers* (Dehghani et al., 2018a) theoretically demonstrates that when a block of middle layers of a fixed transformer model is repeated for a variable number of times, its expressivity can be significantly improved (Saunshi et al., 2025; Merrill & Sabharwal, 2025; Fan et al., 2024). For example, Merrill & Sabharwal (2025) proves that a looped transformer with depth $\Theta(\log n)$ are in $\mathsf{TC}^1$ and thus solve regular language recognition and graph connectivity problems with input context length $n$, which are intractable by LLMs with logarithmic CoT steps. Intuitively, LTs do not decode latents into discrete tokens until the last loop time, enabling implicit conduction of complex reasoning such as planning and searching in the **continuous latent space** and efficient storage of these meaningful intermediate information efficiently for later loops. Taking advantage of a

powerful latent space, scaling the looped depth of the model (even with uniform parameters) can be a more effective way for test-time scaling compared to increasing the length of reasoning steps with explicit CoT in the discrete, finite token space. Unfortunately, despite the strong expressivity, LT scheme is not widely adopted in mainstream LLMs due to the limited practical performance, which we attribute to *the lack of supervision on intermediate states* – the intermediate rollouts are not supervised properly in training, thus LT inference could encounter out-of-distribution (OOD) issues.

On the other hand, diffusion language models (DLMs) (Gong et al., 2024; Nie et al., 2025; Ye et al., 2025) have received considerable attention from researchers in recent years. The **non-autoregressive** nature of DLMs enables *any-order generation*, *self-correction*, and *parallel decoding* capabilities, leading to potentials in stronger expressiveness and superior generation efficiency. State-of-the-art DLMs outperform LLMs in complex or structured reasoning tasks such as Sudoku (Kim et al., 2025) and coding (Gong et al., 2025). There are two main families of DLMs: continuous diffusion models (CDMs) based on SDE or PF–ODE (Gong et al., 2022; Li et al., 2022; Sahoo et al., 2025), and discrete diffusion models (DDMs) based on the continuous-time Markov chain (CTMC) (Austin et al., 2021; Lou et al., 2023; Sahoo et al., 2024; Shi et al., 2024). Continuous DLMs, performing iterative denoising in either embedding space or probability space, emerged earlier but fell far behind AR LLMs in practical performance, until the recent success of discrete DLMs. Intriguingly, discrete diffusions with masked noises tend to outperform those with uniform noises (Amin et al., 2025), at the cost of losing self-correction capabilities. Analogously to LLMs, discrete DLMs also reason in the explicit token space and may partially lose information of previous decoding steps.

In this paper, we conceptually connect all the aforementioned models and algorithms, and propose a new language modeling paradigm that combines the advantages of previous methods. In Section 3, we systematically compare these models from the perspective of **theoretical expressivity** and **practical trainability** (Figure 1). We first show in Section 3.1 that: (i) continuous DLMs are more powerful than their discrete counterparts; (ii) continuous diffusion generalizes looped transformer, which is already partially more expressive than CoT. However, previous heuristics in performance seem to contradict the theoretical expressiveness, which we try to elucidate in Section 3.2 from the perspective of trainability. Looped transformers face OOD issues in inference due to lack of intermediate supervision, while diffusion models supervise states in the whole probability path. We then attribute the insufficient trainability of continuous DLMs to the large decision space, the low-quality embeddings, and combinatorial decoding complexity.

Based on these insights, in Section 4 we propose a new language modeling paradigm named **Coevolutionary Continuous Discrete Diffusion** (CCDD), which combines the advantages of both continuous and discrete diffusion while eliminating the shortcomings via their complementarity. CCDD defines a joint diffusion process on both the discrete state space through CTMC and the continuous probability or embedding space through SDE (Section 4.1). In the reverse process, one denoising model taking the partially noised tokens of both modalities as inputs learns to predict the data distribution in both spaces. Inspired by DiT (Peebles & Xie, 2023), MM-DiT (Esser et al., 2024), and MoE (Shazeer et al., 2017), we design several architectures for joint denoising with various parameter and computation efficiency (Section 4.2). In implementation, we adopt the contextualized embedding space from concurrent pretrained text embedding models such as Qwen3-Embedding (Zhang et al., 2025), which provides rich semnatics modeling joint distributions and injects knowledge from pretrained LMs via implicit representation guidance. CCDD further enjoys great inference flexibility, adaptively balancing between sampling quality (through inference-time scaling with SDE) and efficiency (through few-step sampling with ODE), thanks to improved sampling algorithms including representation classifier-free guidance (CFG). To summarize, CCDD features both the strong expressive power of continuous diffusion and the good trainability of discrete diffusion.

We experimentally validate the effectiveness of CCDD through extensive text modeling tasks, showing that it reduces over $35\%$ perplexity compared with baselines of the same scale on LM1B and OWT dataset, using comparable parameters and FLOPs. Remarkably, our model reveals surprisingly *superior few-step generation quality* compared with discrete baselines: CCDD with even only 8 steps achieves lower generative perplexity compared to MDLM with 256 steps. In complex reasoning tasks such as Sudoku, 3-SAT and countdown, CCDD also surpasses all previous methods including autoregressive models, masked discrete diffusion, and looped transformers. Our code is publicly available at https://github.com/zhouc20/CCDD.

## 2. Preliminary

**Notation.** Let $\Omega = \{1, \ldots, V\}$ be a vocabulary ($|\Omega| = V$) and $L$ the sequence length. Suppose we have the discrete sequence data $x_0 \in \Omega^L$. A fixed encoder $\mathcal{E}$ maps tokens to continuous embeddings $z_0 = \mathcal{E}(x_0) \in \mathbb{R}^{L \times d}$, which can be either one-hot on the simplex $\Delta^{V-1} := \{p \in \mathbb{R}_{\geq 0}^V : \mathbf{1}^\top p = 1\}$ (namely $d = V$) or representations of any pretrained model / LLM – hence, we may use the terms "logits" and "representations" interchangeably. We write $t \in [0, 1]$ for continuous time or $t \in \{1, \ldots, T\}$ for discrete steps, where the latter can also be converted to discretized

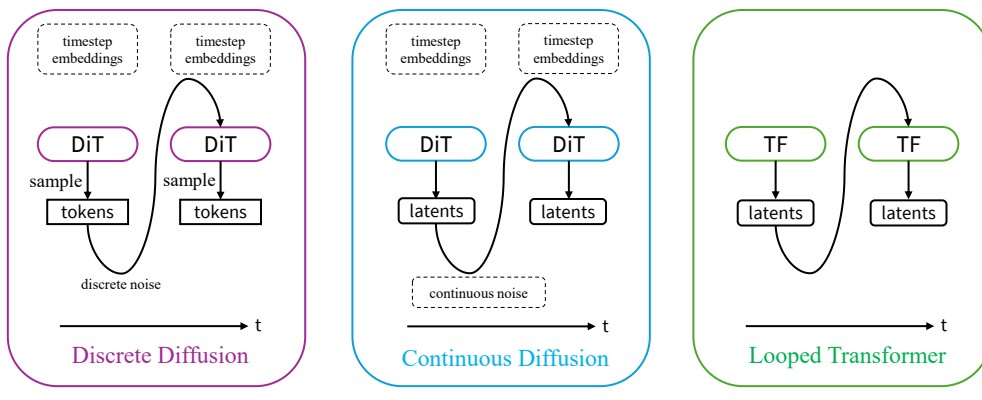

Theoretical Expressivity (Section 3.1): CDM $\succcurlyeq$ DDM    CDM $\succcurlyeq$ LT
Empirical Trainability (Section 3.2): DDM $\succcurlyeq$ CDM $\succcurlyeq$ LT

*Figure 1.* Comparison of theoretical expressiveness and practical trainability of: discrete diffusion (left), continuous diffusion with optional continuous noise (middle), and looped transformer (right).

$\tilde{t} \in [0, 1]$. Denote corrupted variables as $z_t$ and $x_t$, and in Gaussian settings $\epsilon_t \sim \mathcal{N}(0, I)$ denotes the standard noise used to synthesize $z_t$. A *single* time–conditioned network $f_\theta(\cdot, t, \mathrm{cond})$ (typically a transformer-based model) is called at every step/instant, where $\mathrm{cond}$ is the optional condition (often omitted).

**Continuous diffusion.** The forward/noising dynamics for continuous data $z_t \in \mathbb{R}^{L \times d}$ are

$$dz_t = a_t(z_t) \, dt + g_t \, dW_t, \tag{1}$$

with drift $a_t(\cdot)$, scalar (or matrix) diffusion $g_t \geq 0$, and Wiener process $W_t$. The marginals $q_t(z_t)$ satisfy the Fokker–Planck PDE $\partial_t q_t = -\nabla \cdot (a_t q_t) + \frac{1}{2} g_t^2 \Delta q_t$. A standard instance is the variance preserving (VP) schedule: $dz_t = -\frac{1}{2}\beta_t z_t \, dt + \sqrt{\beta_t} \, dW_t$, yielding a closed-form forward marginals:

$$z_t = \alpha_t z_0 + \sigma_t \epsilon_t, \ \ \epsilon_t \sim \mathcal{N}(0, I),$$
$$\alpha_t = \exp\left(-\frac{1}{2}\int_0^t \beta_\tau \, d\tau\right), \ \ \sigma_t = \sqrt{1 - \alpha_t^2}. \tag{2}$$

The reverse process is based on the reverse SDE in (3) or the PF–ODE in (4):

$$dz_t = \left(a_t(z_t) - g_t^2 s_\theta(z_t, t)\right) dt + g_t \, d\bar{W}_t, \tag{3}$$
$$\dot{z}_t = a_t(z_t) - \frac{1}{2} g_t^2 s_\theta(z_t, t). \tag{4}$$

where $s_\theta(\cdot, t) \approx \nabla_z \log q_t(\cdot)$ is produced by $f_\theta$ (up to a known scaling). Under (2), we have closed form sampling update rules as in DDPM (Ho et al., 2020) and DDIM (Song et al., 2020). In practice, common equivalent heads include $\epsilon$-pred: $\epsilon_\theta(z_t, t)$, $x_0$-pred: $\hat{z}_{0,\theta}(z_t, t)$, $v$-pred: $v_\theta(z_t, t)$, which are equivalent up to linear transformations. A typical VP training loss with $\epsilon$-prediction and time-dependent weight $\lambda_{\mathrm{cont}}(t)$ is

$$\mathcal{L}_{\mathrm{cont}} = \mathbb{E}_{t, z_0, \epsilon}\left[\lambda_{\mathrm{cont}}(t) \|\epsilon - \epsilon_\theta(\alpha_t z_0 + \sigma_t \epsilon, t)\|^2\right]. \tag{5}$$

**Discrete diffusion.** For a single token ($L = 1$ for notation), let $q_t \in \Delta^{V-1}$ be the column vector of token marginals. In the forward process, a time-inhomogeneous continuous-time Markov chain (CTMC) with generator $G_t \in \mathbb{R}^{V \times V}$ evolves as follows ($\mathcal{T}$ the normalizing constant):

$$\dot{q}_t = G_t q_t, \ P_{s \to t} = \mathcal{T} e^{(\int_s^t G_\tau \, d\tau)}, \ q_t = P_{0 \to t} q_0. \tag{6}$$

Some common choices of $G_t$ include: (i) *Uniform noise (USDM)*, where state $j$ jumps to any $i \neq j$ uniformly at rate $u_t$, resulting in $(G_t)_{ij} = \frac{u_t}{V-1}$ $(i \neq j)$, $(G_t)_{jj} = -u_t$; (ii) *Masked (absorbing) noise*, where we Augment $\Omega$ with a mask state [MASK]. For any $j \neq$ [MASK], it jumps to the mask state with rate $u_t$, leading to $(G_t)_{[\text{MASK}],j} = u_t$, $(G_t)_{jj} = -u_t$, $(G_t)_{\cdot, [\text{MASK}]} = 0$. The marginal of both processes can be expressed by an interpolation between the clean data and a noise distribution $\pi_t$,

$$q_t(x_t|x_0) = \mathrm{Cat}(\eta_t x_0 + (1 - \eta_t)\pi_t) \tag{7}$$

where $\pi_t = \boldsymbol{m}$ the one-hot vector for [MASK] for the absorbing noise, and $\pi_t = \frac{1}{V}\boldsymbol{1}$ for the uniform noise. For sequences of length $L$, corruptions are exerted independently per-position. In the reverse process, the denoising network predicts the clean data distribution $\mathbf{x}_\theta := \hat{\pi}_\theta(x_0|x_t, t) = \mathrm{softmax}(\ell_\theta(x_t, t))$ where $\ell_\theta$ the output logits. A Bayesian form of posterior is

$$p_\theta(x_s|x_t) = q_{t|s}(x_t|x_s)\frac{q_s(x_s|\hat{\pi}_\theta)}{q_t(x_t|\hat{\pi}_\theta)} \tag{8}$$

The training loss is usually calculated as with weights $\lambda_{\mathrm{disc}}(t, x_t, x_0)$ derived from Rao-Blackwellized likelihood bounds and would be zero for unmasked token in masked diffusion:

$$\mathcal{L}_{\mathrm{disc}} = \mathbb{E}_{t, x_0}\left[\lambda_{\mathrm{disc}}(t, x_t, x_0) \log\langle\hat{\pi}_\theta(x_t, t), x_0\rangle\right] \tag{9}$$

**Looped transformer.** In the standard setting of looped transformer, a *single* $n$-layer transformer ($\Phi_\theta$) block with shared parameters $\theta$ is rolled out adaptive $T$ steps. Let $h_k \in \mathbb{R}^{L \times d}$ be the hidden state after $k$ steps.

$$h_{k+1} = \Phi_\theta(h_k), \qquad k = 0, \dots, T-1. \quad (10)$$

The transformer layers can be either encoder-based (bidirectional attention) or decoder-based (causal attention) with residual connections. A readout $R(h_T) : \mathbb{R}^{L \times d} \to \mathbb{R}^{L \times V}$ yields logits $\ell_{\theta,T} = R(h_T)$ and token samples $x_T \sim \mathrm{Cat}(\mathrm{softmax}(\ell_{\theta,T}))$. Typically, LTs receive supervision on the final outputs using standard cross-entropy loss.

# 3. Theoretical Expressivity and Practical Trainability Analysis

## 3.1. Theoretical Expressivity Analysis

In this subsection, we analyze the theoretical expressivity of CDM, DDM and LT. We assume standard measurability/Lipschitz conditions when needed. Throughout the paper, for expressiveness comparison we assume the *same* architectures and parameter counts in networks with *finite capacity* as in common practice. By default, we consider Transformers (TF) (Vaswani et al., 2017) and Diffusion Transformers (DiT) (Peebles & Xie, 2023), up to slight differences in the first encoding layer and the last decoding layer. Proofs are available in Appendix B.

**Continuous diffusion dominates discrete diffusion.** We first compare the families of trajectory laws and terminal distributions induced by CDM on $\mathcal{Z} := \mathbb{R}^{L \times d}$ and DDM on $\mathcal{X} := \Omega^L$ embedded into $\mathcal{Z}$ via a bijective encoder $\mathcal{E}$. Denote the distribution induced from the reverse SDE (3) as $p_t(z_t) \in \mathcal{P}(\mathcal{Z})$, and $p_t(x_t) \in \mathcal{P}(\mathcal{X})$ produced by the posterior of CTMC (8).

**Definition 3.1** (Trajectory families and embedded discrete family). Define the trajectory family of continuous diffusion $\mathsf{F}_{\mathrm{cont}}(\theta)$, discrete diffusion $F_{\mathrm{disc}}(\theta)$, and the *embedded discrete family* $\widetilde{\mathsf{F}}_{\mathrm{disc}}(\theta) \subset \mathcal{P}(\mathcal{Z})$ (the pushforward by the fixed encoder $\mathcal{E}$) as follows:

$$\mathsf{F}_{\mathrm{cont}}(\theta) := \left\{ \{p_t(z_t)\}_{t \in [0,1]} \right\},$$
$$\mathsf{F}_{\mathrm{disc}}(\theta) := \left\{ \{p_t(x_t)\}_{t \in [0,1]} \right\}, \quad (11)$$
$$\widetilde{\mathsf{F}}_{\mathrm{disc}}(\theta) := \mathcal{E}_\sharp \mathsf{F}_{\mathrm{disc}}(\theta)$$

**Theorem 3.2** (Strict trajectory-level gap). *At any fixed $t \in [0,1]$, we have the following strict inclusion*

$$\widetilde{\mathsf{F}}_{\mathrm{disc}}(\theta) \subsetneq \mathsf{F}_{\mathrm{cont}}(\theta) \subseteq \mathcal{P}(\mathcal{Z}) \quad (12)$$

The key insight here is the fact that the input of denoising network in DDM is always discrete and *supported on a finite*

*set* (Lemma B.6), while the Fokker-Planck equation in CDM would yield *absolutely continuous marginals* (Lemma B.7). The inclusion holds for the entire sampling trajectories so as the terminal distributions. Intuitively, considering both two models operating on the probability simplex – analogously for other embedding spaces given the encoder bijective. The "logits→sample→embed" operation in discrete diffusion sampling loop *quantizes* the cross-step memory into a single token per step, losing access to the full logits. However, the "logits→logits" procedure in continuous diffusion propagates a continuous state, retaining fine-grained uncertainty and historical memory. The discrete scheme imposes a hard finite-support bottleneck with information loss at every step (Lemma B.9), making it strictly dominated by the continuous counterpart producing non-atomic outputs.

*Remark* 3.3 ("Finite-combination" viewpoint). Discrete diffusion operates over convex combinations of finitely many basis states in $\Delta^{V-1}$. This is a strict subset of the continuous family, which admits general smooth densities via (1). Lemma B.9 shows that per-step token sampling discards the full logit geometry, whereas continuous samplers propagate full $z_t$ (ODE deterministically or SDE stochastically) without compulsory quantization at intermediate times.

In addition to the functional class inclusion, we also highlight that continuous diffusion is capable of modeling the joint distribution over tokens and internally supports ODE sampling, while discrete diffusion models the token marginals and can only sample with SDE. Therefore, CDM has huge potentials in global planning and few-step generation, which is verified through our experiments in Section 5.

**Continuous diffusion generalizes looped transformer.** It is known that looped transformers are already partially more expressive than CoT (Saunshi et al., 2025; Merrill & Sabharwal, 2025; Fan et al., 2024). We now further show that a continuous diffusion, in principle, can *simulate* any looped transformer with the same architecture and parameter count, hence is at least as expressive as the powerful looped transformer (and potentially even more expressive).

**Proposition 3.4** (Continuous diffusion sampler simulates looped rollouts). *Fix any looped transformer $\Phi_\theta$ and roll out times $T \in \mathbb{N}$, there exists a continuous diffusion sampler for the reverse PF–ODE by the explicit Euler method with step size $1/T$ that exactly reproduces the looped rollout.*

The crucial intuition is one can always construct a reverse PF–ODE with grid outpoints matching the looped transformer roll outs, and is realizable with a denoising network with the same architecture and parameter budget as the looped transformer. The contrary does not hold: a deterministic looped rollout cannot simulate any non-degenerate stochastic path produced by a continuous diffusion with $g_t > 0$ in the reverse SDE (Theorem B.14).

*Table 1.* Comparison between generation space of continuous diffusion.

|  | **Simplex** $\Delta^{V-1}$ | **Token-wise** $\mathbb{R}^d$ | **Contextualized** $\mathbb{R}^d$ |
|---|---|---|---|
| Dimensionality | $V-1$ (high) | $d \leq V$ (often $\ll V$) | $d \leq V$ (often $\ll V$) |
| Geometry | Constrained manifold | Euclidean; codebook cells | Euclidean; contextual manifold |
| Target smoothness | Low (near vertices) | Atomic, non-smooth | Higher (good embedding models) |
| Calibration | Natural | Requires decoder | Requires decoder (context) |
| Expressivity (terminal) | Baseline | $\not\succ$simplex (Prop. E.1) | $\geq$ simplex if decoder strong |
| Decoding ambiguity | Low | Medium (NN/energy) | High if not sufficient |
| Optimization | Hard (constraints) | Boundary brittle | Complex but smoother targets |

## 3.2. Empirical Performance and Practical Trainability

Intriguingly, despite their strong theoretical expressiveness, previous looped transformers tend to exhibit limited empirical performance compared with SOTA LLMs. Meanwhile, continuous DLMs typically underperform their discrete counterparts, contradicting to the expressivity inclusion. In this subsection, we analyze these empirical observations through the lens of practical trainability.

**Advantages of intermediate supervision.** We point out a drawback in classical looped transformer training: they are typically trained as standard transformers (i.e., depth $T = 1$), or trained with a fixed depth and only supervised on the last roll out. Consequently, LT would encounter out-of-distribution (OOD) problems when rolled out with a different time from training, since supervisions on the intermediate states of these depths are never received.

Fortunately, continuous diffusion models naturally address this problem. During training, all continuous time instances (or sufficiently dense discrete timesteps) would be sampled and supervised by denoising loss, so the model is able to model all intermediate timesteps along the probability path. The progressively denoising parameterization also enables flexible number of function evaluations (NFEs) or diffusion timesteps in inference, which is hard for looped transformers without sophisticated special design. Combining the advantage brought by *intermediate supervision* and the theoretical expressivity inclusion in Proposition 3.4, we conclude that instead of LTs, one can actually train CDMs which are expressive and easier to optimize.

**Limitations in trainability of continuous diffusion.** While theoretically expressive, previous CDMs typically underperform their discrete counterparts and AR LLMs in practice, necessitating us to rethink the reasons behind. In addition to the gap in engineering efforts, we argue that there are fundamental challenges of existing continuous diffusion: the larger decision space, the combinatorial complexity in decoding, and the deficient representation space.

We summarize three generation spaces for CDMs in Table 1 and leave more rigorous definitions and detailed discussions

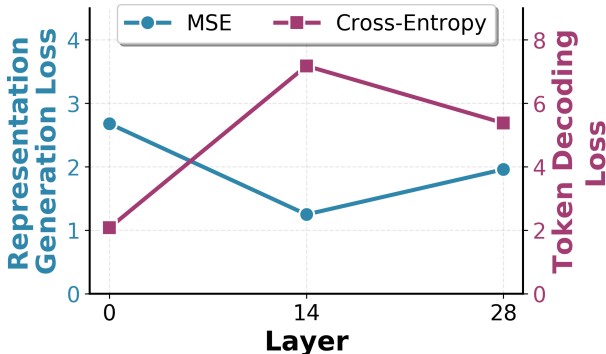

*Figure 2.* Comparison of validation losses when using representations from different layers of Qwen3-Embedding-0.6B as the latent spaces for CDMs.

to Appendix E. In fact, all of them adhere *larger decision spaces* compared with DDMs, which brings expressiveness gain yet incur harder optimization. In particular, the probability simplex adopted by (Han et al., 2022; Sahoo et al., 2025) is *high-dimensional* with potential hard constraints. Token-wise embedding space is the most common choice of early CDMs (Gong et al., 2022; Li et al., 2022), which we argue is a *deficient representation space* as it is not more expressive than the simplex with dimension $d \leq V$ (Proposition E.1). Moreover, the generation target is the atomic codebook representations (essentially a set in $\mathbb{R}^d$), posing difficulty for a CDM to generate. By contrast, contextualized embeddings (where token embeddings depends on the sequence contexts, e.g. hidden features in LLMs) provide more semantic information of the contexts and serve as a smoother generation target – especially those high-quality representations from pretrained LLMs. However, the complicated and ambiguous contextualized embeddings present more difficulties in *decoding featuring combinatorial complexity*. The analysis above is supported by quantitative experimental results demonstrated in Figure 2: utilizing the first layer of Qwen3-Embedding as the generation space (which produces essentially token-wise embeddings) results in the smallest cross-entropy but the largest generation MSE, while the last layer (which gives contextualized embeddings) lead to moderately small MSE and larger classification loss.

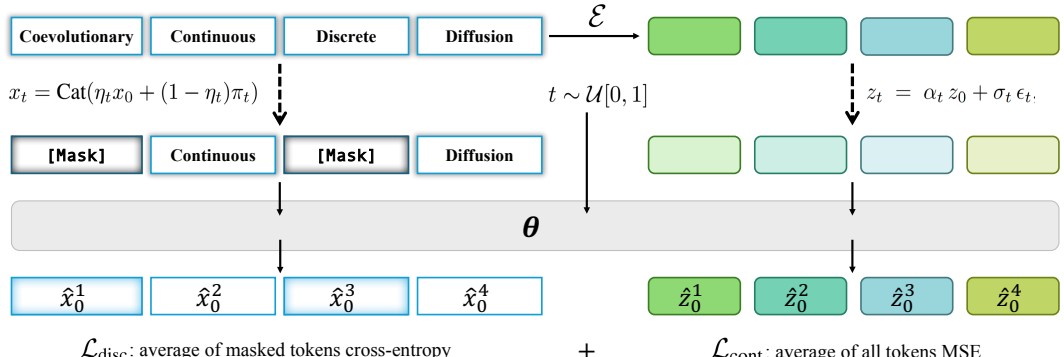

*Figure 3.* Framework of Coevolutionary Continuous Discrete Diffusion.

# 4. Coevolutionary Continuous Discrete Diffusion

Based on the insights in Section 3, we aim to **bridge the gap between theoretical expressivity and practical learnability**. In this section, we overcome the expressivity-trainability dilemma by combining the continuous representation space with discrete diffusion. Remarkably, the discrete state reduces the uncertainty and complexity of input space, making optimization and decoding of the model easier. The continuous space enlarges expressivity upper bound, and representations from well-pretrained LLMs significantly improve the generation quality.

## 4.1. Joint Continuous-Discrete Diffusion

Now we introduce Coevolutionary Continuous Discrete Diffusion (CCDD), a diffusion model on the joint of discrete and continuous space $\mathcal{X} \times \mathcal{Z}$ (Figure 3). In particular, we consider a *joint corruption process* $(x_t, z_t) \in \mathcal{X} \times \mathcal{Z}$ that applies noise individually to each component, and a denoising process that *conditions* on both $(x_t, z_t)$ but updates each component with its own modality-specific rule, i.e., both forward and backward process are parametrically factored.

**Forward process.** Let the forward law be the *product* of a CTMC on $\mathcal{X}$ and an SDE on $\mathcal{Z}$, both possibly time-inhomogeneous, and independent *conditional on* $(x_0, z_0)$, which gives the factored conditional forward kernels:

$$q_t(x_t, z_t \mid x_0, z_0) = q_t^{\text{disc}}(x_t \mid x_0) \, q_t^{\text{cont}}(z_t \mid z_0). \quad (13)$$

For instance, considering the representative forward process for continuous diffusion in (2) and discrete diffusion in (7), the corresponding $(x_t, z_t)$ follows:

$$z_t \sim \mathcal{N}(\alpha_t z_0, \sigma_t^2 I), \quad x_t \sim \text{Cat}(\eta_t x_0 + (1 - \eta_t)\pi_t). \quad (14)$$

**Reverse process.** A single time–conditioned network $f_\theta(\cdot, t)$ receives inputs $(x_t, z_t)$ and outputs modality-specific heads. We adopt the following *factored* reverse update at a time step $t \to s$ $(s < t)$, conditioned on the *multimodal pair* $(x_t, z_t)$:

$$p_\theta(x_s, z_s \mid x_t, z_t) = p_\theta^{\text{disc}}(x_s \mid x_t, z_t) \, p_\theta^{\text{cont}}(z_s \mid x_t, z_t). \quad (15)$$

For simplicity, we illustrate with $x_0$-prediction, while other standard parameterizations such as $\epsilon$-prediction and $v$-prediction are equivalent. While the estimation for each modality depends on both states, i.e., $\hat{x}_{0,\theta} = \hat{x}_{0,\theta}(x_t, z_t, t)$, $\hat{z}_{0,\theta} = \hat{z}_{0,\theta}(x_t, z_t, t)$, the following updates are carried on separately based on the original rules such as DDPM (3) or DDIM (4) for $z_t$ and the Bayes posterior (8) for $x_t$; see Algorithm 2 for algorithmic description.

Intuitively, this reverse process combines the *radical* discrete decoder with high confidence, and the *conservative* continuous decoder with full uncertainty information. Thanks to the continuous component, CCDD is able to preserve full semantics in previous denoising steps and later leverage these historical information, which would mostly be discarded by masked DDM. CCDD is also capable of striking the balance between inference efficiency through few-step ODE sampling, and generation quality through test-time scaling with SDE.

Based on the established ELBOs for continuous and discrete diffusion, we calculate the loss for two modalities according to (5) and (9), respectively. As illustrated in Algorithm 1, the CCDD training loss is a (weighted) sum of two losses:

$$\mathcal{L}_{\text{CCDD}} = \gamma_{\text{cont}} \cdot \mathcal{L}_{\text{cont}} + \gamma_{\text{disc}} \cdot \mathcal{L}_{\text{disc}} \quad (16)$$

*Remark* 4.1 (Conditioning vs. factorization). Although (15) factorizes the *kernel* at each step, each factor is allowed to depend on *both* inputs $(x_t, z_t)$. Thus cross-modal coupling is injected via conditioning: $x$-updates see $z_t$ and vice versa. In other words, the factorization provides an efficient parameterization without making $x_t$ and $z_t$ to be independent in the reverse process. In fact, the factored forward processes admit semigroups (Theorem B.18), and the factored reverse kernels with sufficiently small steps yield aymptotically the same expressivity as fully coupled kernels (Theorem B.19).

*Table 2.* Validation perplexity on LM1B. We use Qwen3-Embedding-0.6B as the continuous generation space for CCDD, and reimplement the baselines with the same Qwen-2 tokenizer. CCDD-MDiT with the same number of parameters of FLOPs significantly outperforms the discrete-only MDLM baseline, reducing over 25% perplexity. CCDD-MoEDiT and MMDiT further improve the performance.

| Model | Train. toks. | # params. | Validation PPL ($\downarrow$) |
|---|---|---|---|
| MDLM (Sahoo et al., 2024) (reimpl.) | 33B | 92.1M | $\leq 39.17$ |
| CCDD-MDiT w/ Qwen3 (ours) | 33B | 92.1M | $\leq 29.22$ |
| CCDD-MoEDiT w/ Qwen3 (ours) | 33B | 104.0M | $\leq 28.50$ |
| CCDD-MMDiT w/ Qwen3 (ours) | 33B | 216.2M | $\leq \mathbf{25.76}$ |

### 4.2. Implementation Techniques

**Architecture design.** Based on DiT (Peebles & Xie, 2023), MM-DiT (Esser et al., 2024), and MoE (Shazeer et al., 2017), we design several effective architectures for joint denoising with various parameters and complexities (Figure 5). In particular, MDiT introduces no additional parameters or FLOPs in non-embedding layers; MMDiT and MoEDiT increase parameters and FLOPs in different manners with significantly better performance. More details are available in Appendix C.1.

**Selection of continuous space: representation learning perspective.** Based on the above analysis, we select contextualized embedding space obtained from pretrained LLM-based text encoders, such as Qwen3-Embedding (Zhang et al., 2025). The *contextualized* embeddings provides rich sequence-level semantics and is easier to generate. Moreover, from a *representation learning perspective*, the high-quality latents serve as representation regularization that accelerates the convergence of training (Yu et al., 2024; Wang et al., 2025), and the high-level conditioning or guidance in inference (Li et al., 2024a; Kouzelis et al., 2025).

**Classifier-free guidance.** The continuous representations can be viewed as self-generated representation guidance for discrete token generation. Analogously to classifier-free guidance (CFG) (Ho & Salimans, 2022), we treat the dual-modality forward as the conditional model (with output $\text{logits}_c$), and the discrete-only forward as the unconditional model (with output $\text{logits}_\phi$). In training, we randomly zero-in and zero-out all continuous token embeddings with probability $p_{\text{drop}}$, hence the model forwards with only discrete states. In sampling, the logits per-step with CFG are computed as $\text{logits} = w \cdot \text{logits}_c + (1 - w) \cdot \text{logits}_\phi$ with the guidance scale $w$.

To summarize, CCDD is a novel language modeling regime that combines multimodal spaces to generate unimodal texts. CCDD generalizes CDM, DDM and LT, featuring high expressivity upper bound; meanwhile, the learnability is also improved through synergy: the discrete component provides capabilities for decoding, and the continuous component benefits from representation learning.

## 5. Experiments

**Experimental setup.** We pretrain our models on the widely used One Billion Words Dataset (LM1B) (Chelba et al., 2013) and OpenWebText (OWT) (Gokaslan & Cohen, 2019) dataset, following most settings in prior work (von Rütte et al., 2025; Shi et al., 2024; Lou et al., 2023; Sahoo et al., 2024). For LM1B, we use the standard split, and train models using sequence length $L = 128$ with sentence packing. For OWT, following (Sahoo et al., 2024; von Rütte et al., 2025), we reserve the last 100K documents as the validation set, and adopt sequence length $L = 512$ with sentence packing. Instead of bert-base-uncased tokenizer on LM1B and GPT-2 (Radford et al., 2019) tokenizer on OWT, for both datasets we use GPT-2 tokenizer when train CCDD with RoBERTa (Liu et al., 2019), and use Qwen-2 (Yang et al., 2024a) tokenizer (also adopted by Qwen-3 series of models) when leverage Qwen3-Embedding (Zhang et al., 2025) representations. Notably, perplexity calculated with different vocabulary sizes are not comparable: Qwen-2 tokenizer has approximately $3\times$ vocabulary size compared with GPT-2, naturally resulting in larger ELBO and perplexity – we thus reproduce the baselines with the same tokenizer. We develop our transformer architectures MDiT, MMDiT, and MoEDiT (detailed in Appendix C.1) based on (Lou et al., 2023) with the same configurations when plausible, which augments DiT (Peebles & Xie, 2023) with rotary embeddings (Su et al., 2024). All models are trained for 1M steps with batch size 512 on both datasets, corresponding to 33B tokens on LM1B and 131B tokens on OWT. We also evaluation CCDD on downstream benchmarks and complex reasoning tasks. More experimental setup, implementation details and additional results are deferred to Appendix D.

**Main results.** CCDD consistently outperforms discrete baselines on both benchmark, with comparable parameter and computation conjectures.

The results on LM1B are reported in Table 2. With the help of the powerful Qwen3-Embedding representations, CCDD-MDiT reduces validation perplexity by over 25% compared with MDLM baseline using the same number of parameters. Moreover, it takes MDLM 1000k iterations to achieve the 39.17 perplexity, while CCDD needs only 40k

*Table 3.* Validation perplexity on OWT with Qwen-2 tokenizer. CCDD is trained with Qwen3-Embedding-0.6B embeddings.

| Model | Train. toks. | # params. | Validation PPL ($\downarrow$) |
|---|---|---|---|
| MDLM (reimpl.) | 131B | 92.1M | $\leq 33.78$ |
| CCDD-MDiT w/Qwen3 (ours) | 131B | 92.1M | $\leq 29.18$ |
| CCDD-MoEDiT w/Qwen3 (ours) | 131B | 104.0M | $\leq \mathbf{21.90}$ |
| CCDD-MMDiT w/Qwen3 (ours) | 131B | 124.0M | $\leq 26.59$ |

*Table 4.* Validation perplexity on OWT with GPT-2 tokenizer. CCDD is trained with the simple RoBERTa-base or Qwen3-Embedding-0.6B embeddings, but still outperforms discrete baselines.

| Model | Train. toks. | # params. | Validation PPL ($\downarrow$) |
|---|---|---|---|
| GPT2 (Radford et al., 2019)[†] | unk. | 117M | 23.40 |
| Llama110M (retrain.)[†] | 262B | 110M | 16.11 |
| SEDD (Lou et al., 2023)[†] | 262B | 92.1M | $\leq 24.10$ |
| MDLM (Sahoo et al., 2024) (reimpl.) | 131B | 92.1M | $\leq 27.39$ |
| GIDD+ (von Rütte et al., 2025) (reimpl.) | 131B | 92.1M | $\leq 25.82$ |
| CCDD-MoEDiT w/RoBERTa (ours) | 131B | 104.0M | $\leq 24.56$ |

iterations, resulting in a $25\times$ **training acceleration**. Scaling the number of parameters via architectural improvement further enhances the performance.

Shown in Table 3 and Table 4 are the results on the more challenging OWT dataset trained with Qwen-2 tokenizer and GPT-2 tokenizer respecptively. Table 4 demonstrates that even the simple RoBERTa embeddings could benefit CCDD training. By switching to the well-pretrained Qwen3-Embedding space and scaling the parameters, CCDD reveals significantly larger advantages (Table 3), reducing over $35\%$ validation perplexity with comparable parameters.

**Classifier-free guidance.** To measure the inference-time flexibility, we report the generative NLLs of CCDD samples with inference-time CFG in Table 5. We generate 256 samples with sequence length 512 using 512 denoising steps parameterized by DDPM and MDLM reverse process. We use GPT2-Large as the reference model, and the generative perplexity is calculated as the exponential function of the NLL. With either discrete-only forward ($w = 1$) or standard joint forward ($w = 0$), CCDD has superior performance, while CFG further improves the quality, verifying the advantages of latent reasoning.

**Few-step generation performance.** CCDD has superior few-step generation performance, thanks to the capability of modeling joint distribution and global latent reasoning of the continuous component. We evaluate generative perplexity using the configuration above, but with various denoising steps (Figure 4). CCDD with only 8 steps outperforms MDLM with 256 steps, a remarkable $16\times$ acceleration.

*Table 5.* Generative NLLs of CCDD with Qwen3-Embedding pretraining on OWT with Qwen-2 tokenizer using CFG.

| Model | $w$ | Gen. NLL ($\downarrow$) |
|---|---|---|
| MDLM | - | 9.19 |
| CCDD-MoEDiT | 0.0 | 9.06 |
| CCDD-MoEDiT | 1.0 | 8.38 |
| CCDD-MoEDiT | 1.5 | **8.25** |

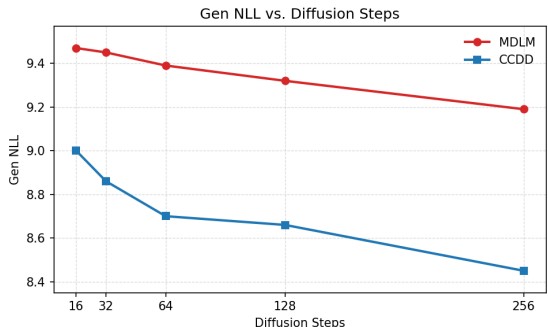

*Figure 4.* Few-step generative NLL with Qwen-2 tokenizer.

**Ablation studies.** As shown in Figure 2, we compare the validation losses when using representations from different layers of Qwen3-Embedding-0.6B as the latent spaces for continuous diffusion models. The losses consist of two components: (i) the representation MSE loss, which we use to measure the difficulty in generating the target representations; and (ii) the token decoding cross-entropy loss, which measures the difficulty in decoding the generated latents into discrete tokens. All the representations are normalized,

*Table 6.* Test accuracy on complex reasoning datasets: Sudoku, 3-SAT, and Countdown.

| Model | Size | Depth $T$ | Sudoku | 3-SAT | Countdown |
|---|---|---|---|---|---|
| GPT2 scratch | 6M | Seq. Len. | 16.2 | 73.1 | 31.9 |
| GPT2 scratch | 303M | Seq. Len. | 19.4 | - | 41.3 |
| Llama | 7B | Seq. Len. | 27.1 | - | 41.1 |
| Llama | 13B | Seq. Len. | 33.8 | - | 51.1 |
| MDM | 6M | 2 | 88.6 | 39.2 | - |
| MDM | 6M | 20 | 99.9 | 87.0 | 52.0 |
| LT | 6M | 2 | **100.0** | 91.3 | 60.6 |
| LT | 6M | 3 | **100.0** | - | 68.2 |
| CCDD (ours) | 6M | 2 | **100.0** | **91.9** | 67.8 |
| CCDD (ours) | 6M | 3 | **100.0** | - | **73.7** |

with a hidden dimension of 32. All models leverage the DiT architectures and are trained on LM1B for 500k steps with same configurations. The results validate our hypothesis: the 0-th layer (token-wise) corresponds to the smallest token loss but the largest representation loss, while the 28-th layer (contextualized) admits moderate losses, striking a balance between the generativity and decodability. We refer readers to Appendix D for more ablation studies.

**Complex reasoning: Sudoku, SAT, and Countdown.** For complex mathematical reasoning tasks, we select Sudoku, SAT and Countdown. We adopt the datasets and experimental settings in (Ye et al., 2024a). Details of the tasks and datasets are available in Section D.4. By default, we train 6M models of different kinds from scratch, keeping hyperparameters consistent with (Ye et al., 2024a).

As shown in Table 6, the results on all three datasets consistently show that while MDM is more powerful than AR models (including the large pretrained LLaMA), looped transformer (LT) with even 2 looped depth have better performance, and *CCDD with just two steps is sufficient to beat all models*. The empirical observations are consistent with our theory: LT is expressive through latent reasoning, and scaling depth is more effective than scaling CoT steps as in AR or increasing discrete diffusion steps in MDM. The results also validate that CCDD is at least as expressive as the better one within MDM and LT, and the joint design brings even further performance gain. Therefore, CCDD benefit from both any-order generation capability of MDM, and the strong papalism/deep effective depth of LT. Furthermore, CCDD is also the most efficient one: with the same architectures and parameter counts, it features more efficient training than LT who needs to roll out all depths in training, and fewer sampling steps suffices compared with MDM.

## 6. Conclusion

The contributions of the paper are sumarized as follows. Theoretically, we systematically analyze mainstream language modeling regimes through the lens of expressivity and trainability. We conclude that under same computation conjecture continuous diffusion dominates discrete diffusion, while being able to simulate looped transformer. However, although CLM overcomes the OOD problem of LT, it still lacks trainability due to several fundamental limitations. Methodologically, we introduce Coevolutionary Continuous Discrete Diffusion (CCDD), which defines a joint diffusion process on both continuous and discrete space and levarages a single model to jointly denoise. CCDD is a novel language modeling scheme that retains both strong expressivity and trainability. We also present effective architectures as well as advanced training and sampling techniques. Experimentally, pretrained CCDD on real-world datasets LM1B and OWT reveals significantly lower validation perplexity compared with baselines, and superior performance in downstream benchmarks and complex reasoning datasets further validates the strength of latent reasoning.

## Impact Statement

This paper presents work whose goal is to advance the field of Machine Learning, Language Generation and Reasoning. There are many potential societal consequences of our work, none which we feel must be specifically highlighted here.

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

# A. Related Work

**Looped transformers and latent CoT.**   Recent research has explored how enable a model with a fixed number of layers to think deeper about problems through architectural design or specialized training, effectively simulating a deeper transformer (Zhu et al., 2025b). A fundamental strategy in this direction involves loop-based architectures (Dehghani et al., 2018b; Mohtashami et al., 2023; Bae et al., 2024). For instance, Fan et al. (2024) show that standard fixed-depth transformers struggle in length generalization which can be significantly improved by looped transformers. Saunshi et al. (2025) first proves that a $T$-depth looped transformer can implicitly generate latent thoughts and can simulate $T$ steps of CoT reasoning under mild assumptions. Furthermore, Merrill & Sabharwal (2025) shows that a looped transformer with depth $\Theta(\log n)$ can solve regular language recognition and graph connectivity with arbitrary input context length $n$, which is intractable by LLMs with logarithmic CoT steps. Mixture of Recursion (Bae et al., 2025) is the recent looped transformer that practically works, which adaptively adjust the looping depth for tokens.

In contrast to architectural recurrence, which necessitates explicit structural changes, an alternative known as *continuous chain-of-thought (continuous CoT)* achieves comparable computational advantages through specialized training of standard transformer models (Hao et al., 2024; Shen et al., 2025; Cheng & Van Durme, 2024; Wang et al., 2024). A representative is COCONUT (Hao et al., 2024), which operates on the continuous token space instead of using recurrent parameters. COCONUT directly treats the last hidden state of previous tokens as reasoning tokens for CoT reasoning, allowing them to explore multiple reasoning paths simultaneously, akin to breadth-first search, without being constrained to natural language tokens (Gozeten et al., 2025; Zhu et al., 2025a). Continuous CoT outperforms standard discrete CoT in certain reasoning tasks demanding parallel searching and multiple reasoning paths, yet still falls behind in general tasks. Notably, latent CoT could be simulated by continuous diffusion with diffusion forcing (Chen et al., 2024), and we focus on looped transformer in the main context for clarity.

**Diffusion language models.**   Diffusion language models emerge as a new paradigm that reformulates text generation as an iterative denoising process, enabling complex reasoning by leveraging full-sequence context. This paradigm primarily includes *Masked Diffusion Models (MDMs)*, which are a type of DDMs, and *Embedding-based Diffusion Models (EDMs)*, a subset of CDMs.

MDMs operate on discrete tokens, starting from a masked sequence and refining tokens simultaneously using bidirectional context. Early foundational work includes D3PM (Austin et al., 2021) and SEDD (Lou et al., 2023), which introduced discrete transition processes and score matching losses. Subsequent methods (Ou et al., 2024; Sahoo et al., 2024; Shi et al., 2024) streamlined training through hybrid masked losses, facilitating the conversion of encoder models like BERT into generative reasoners. The iterative unmasking process inherent in MDMs supports sophisticated reasoning capabilities, such as iterative refinement (Du et al., 2024) and reverse-order reasoning (Nie et al., 2025). The framework has also been integrated with chain-of-thought reasoning (Ye et al., 2024b), demonstrating strong performance in tasks requiring parallel context and systematic refinement. Similar algorithms are proposed from the flow matching perspective (Gat et al., 2024). Additional to mask noises, some work try to leverage uniform noises which tend to have worse performance (von Rütte et al., 2025; Shaul et al., 2024). Another series of work extend intermediate states by introducing partially noised states in between mask and clean tokens (Zhou et al., 2025; Chao et al., 2025), for example, HDLM (Zhou et al., 2025) leverages hierarchies of semantics for each token, where lower-level detailed tokens are noised into higher-level abstract tokens (such as cluster tokens) in the forward process, and the model progressively denoises by predicting the next semantic scale in the reverse process. MDM-Prime (Chao et al., 2025) extends the masking state by converting each word token into several subtokens and gradually mask the subtokens. Notably, these methods still operate in the discrete state spaces.

In contrast, EDMs perform diffusion in a continuous embedding space. EDM research focused on controllable generation (Li et al., 2022) and sequence-to-sequence tasks (Dieleman et al., 2022; Mahabadi et al., 2023; Gong et al., 2022), with Plaid (Gulrajani & Hashimoto, 2023) later establishing empirical scaling laws that significantly narrowed the efficiency gap with autoregressive models. The framework was further extended by DoT-Plaid (Ye et al., 2024b), which generalized chain-of-thought reasoning to EDMs, leveraging iterative latent refinement for improved coherence and mathematical reasoning. There are also a few continuous diffusion models operating on the logit space (Han et al., 2022; Sahoo et al., 2025).

Some previous work have noticed the potential of multimodal generation integrating continuous and discrete diffusion, with applications to text-image joint generation (Rojas et al., 2025) and protein sequence-structure co-design (Campbell et al., 2024). DUO (Sahoo et al., 2025) tries to connect two types of diffusion models via marginal matching, and apply distillation

tricks for continuous diffusion to discrete text diffusion. In the context of diffusion language models, Diffuseq-v2 (Gong et al., 2023) also tries to bridge continuous and discrete generation process by introducing a soft absorbing state. In comparison, our work generalize these results, propose a more principled method for joint continuous-discrete modeling, and provide systematic analysis on expressiveness and trainability. We also practically showcase that combining continuous and discrete models to benefit each other, and the powerful latent reasoning could significantly improve the expressivity and performance in complex reasoning tasks such as Sudoku.

**Representation learning for diffusion models.**   Recent advances in representation-enhanced diffusion model training show that high-quality representations from pretrained models could benefit the training efficiency and sampling quality of diffusion models through flexible ways (Li et al., 2024a;b; Yu et al., 2024; Wang et al., 2025; Kouzelis et al., 2025). In particular, RCG (Li et al., 2024a) and GeoRCG (Li et al., 2024b) adopt two-stage generation processes where a representation generator first samples high-level features which serve as the conditions for the second-stage image or molecule generation. In contrast, REPA (Yu et al., 2024) is a training-time technique that aligns the internal features of diffusion models with external pretrained representations, thereby accelerating the training procedure. REED (Wang et al., 2025) unifies RCG and REPA from a theoretical perspective, and generalizes these methods by leveraging multi-modal representations and improved training curriculum. Furthermore, ReDi (Kouzelis et al., 2025) demonstrates that generating images and their representations at the same time also boosts generation quality of diffusion models.

# B. Omitted Proof

## B.1. Theoretical Expressivity Analysis

This subsection provides proof for Section 3.1 in the main text and presents additional results. We assume standard measurability/Lipschitz conditions when needed.

First we start with the concepts of circuit-complexity classes $\mathsf{TC}^0$ and $\mathsf{TC}^1$

**Definition B.1** ($\mathsf{TC}^0$). A language $L$ is in $\mathsf{TC}^0$ if it is decided by a family of Boolean circuits $\{C_n\}$ such that: (i) $C_n$ has polynomial size; (ii) $C_n$ has *constant depth* $O(1)$; (iii) gates are NOT, AND, OR, and unbounded-fan-in threshold gates.

**Definition B.2** ($\mathsf{TC}^1$). A language $L$ is in $\mathsf{TC}^1$ if it is decided by a family of Boolean circuits $\{C_n\}$ such that: (i) $C_n$ has polynomial size; (ii) $C_n$ has *logarithmic depth* $O(\log n)$; (iii) gates are NOT, AND, OR, and unbounded-fan-in threshold gates.

Some basic containments are:
$$\mathsf{AC}^0 \subsetneq \mathsf{TC}^0 \subseteq \mathsf{TC}^1 \subseteq \mathsf{NC}^2 \subseteq \mathsf{P}. \tag{17}$$

The intuition is that $\mathsf{TC}^0$ computes addition, multiplication, majority, parity, etc. in constant depth. $\mathsf{TC}^1$ allows more power via logarithmic-depth threshold circuits, including division and iterated arithmetic. Our results show that analogously to LT, CDM and CCDD can be in class $\mathsf{TC}^1$, showing the advantage of latent reasoning and continuous DLM.

### B.1.1. CONTINUOUS DIFFUSION DOMINATES DISCRETE DIFFUSION

We now give rigorous definitions of decision space and representation space of CDM and DDM.

**Definition B.3** (Decision Space). Given any generation task and the context cond, the decision space is just the set of all possible outputs by the model parameterized with $\theta$. For a single token being generated, the decision space of a discrete diffusion is clearly $\Omega$, and that of a continuous diffusion is either $\Delta^{V-1}$ or $\mathbb{R}^d$ as already stated in Table 1. For multiple tokens with length $L$, the definition decision spaces can be easily extended to the combinations of single token decision spaces, i.e., $\Omega^L$ for discrete diffusion and $(\Delta^{V-1})^L$ or $(\mathbb{R}^d)^L$ for continuous diffusion.

It is straightforward that the decision space of continuous diffusion is larger than the discrete diffusion: the former is absolutely continuous w.r.t. Lebesgue measure on $\mathbb{R}^{L \times d}$ (See Lemma 4), while the latter is finitely supported by a set (See Lemma 3). Thus the continuous diffusion is naturally harder to be learned by a neural network under same capacity.

**Definition B.4** (Representation Space). The representation space is where the generation targets of a (latent) continuous diffusion lie in, determined by the pre-defined by the encoder $\mathcal{E}$, namely $\{\mathcal{E}(x)|x \in \Omega^L, L \in \mathbb{N}\}$.

The encoder, as already explained in the preliminary section and Section 3.2, can be either the mapping to the probability simplex, or token-wise embedding defined by a codebook (such as a nn.Embedding layer), or a contextualized mapping

through any pretrained LLMs or sentence embedding models (e.g., RoBERTa or Qwen3-Embedding used in the experiments). This is similar to image (latent) diffusion models: the representation space is usually defined by a pretrained VAE or an unsupervised learning model such as DINO.

**Assumption B.5** (Regularity for continuous diffusion). We assume $g_t > 0$ on a set of times of positive measure in $[0, 1]$, and $a_t$ is such that the Fokker–Planck equation is well-posed and yields absolutely continuous marginals for $t > 0$ when starting from a distribution with a density or from any point mass convolved with the Gaussian noise of (1).

**Lemma B.6** (Embedded discrete trajectories are finitely supported at each $t$). *Fix any $t \in [0, 1]$. For any $\{p_t\}_{t \in [0,1]} \in$ $\mathsf{F}_{\mathrm{disc}}(\theta)$, the embedded marginal $q_t := \mathcal{E}_{\sharp} p_t \in \widetilde{\mathsf{F}}_{\mathrm{disc}}(\theta)$ is supported on a* finite *set in $\mathbb{R}^{L \times d}$. In particular, if $\mathcal{E}$ is one-hot or any fixed finite codebook, then $q_t$ is a finite mixture of Dirac masses in $\mathbb{R}^{L \times d}$.*

*Proof.* For any $t$, $p_t$ is a probability vector over the finite set $\mathcal{X} = \Omega^L$ (size $V^L$). Hence $\mathrm{supp}(p_t) \subseteq \mathcal{X}$ is finite. The encoder $\mathcal{E} : \mathcal{X} \to \mathbb{R}^{L \times d}$ maps each $x \in \mathcal{X}$ to a single point $\mathcal{E}(x)$, and thus the pushforward $q_t(B) = p_t(\mathcal{E}^{-1}(B))$ is supported on the finite set $\{\mathcal{E}(x) : x \in \mathcal{X}\}$. Therefore $q_t$ is a finite atomic measure in $\mathbb{R}^{L \times d}$. □

**Lemma B.7** (Continuous diffusion produces absolutely continuous marginals). *Under Assumption B.5, for any $t > 0$, the marginal $q_t \in \mathcal{P}(\mathbb{R}^{L \times d})$ arising from (1) is absolutely continuous w.r.t. the Lebesgue measure on $\mathbb{R}^{L \times d}$. In the VP case (2), $z_t = \alpha_t z_0 + \sigma_t \epsilon$ with $\sigma_t > 0$ implies $q_t$ is a Gaussian smoothing of the law of $z_0$, thus absolutely continuous.*

*Proof.* With $g_t > 0$ on a set of positive measure and standard regularity on $a_t$, the Fokker–Planck operator is (hypo)elliptic on $\mathbb{R}^{L \times d}$. Starting from any initial distribution with a density (or from a point mass, which immediately becomes smooth by Gaussian convolution when $g_t > 0$), the solution $q_t$ admits a density for any $t > 0$. In the VP instance (2), $z_t$ is an affine transformation of $z_0$ plus independent Gaussian noise with variance $\sigma_t^2 I$, hence $q_t$ is the convolution of the law of $\alpha_t z_0$ with a non-degenerate Gaussian, which is absolutely continuous. □

**Theorem B.8** (Strict trajectory-level gap, Theorem 3.2 in main text). *At any fixed $t \in (0, 1]$, we have the following strict inclusion*

$$\widetilde{\mathsf{F}}_{\mathrm{disc}}(\theta) \subsetneq \mathsf{F}_{\mathrm{cont}}(\theta) \quad \text{as subsets of } \mathcal{P}(\mathbb{R}^{L \times d}) \tag{18}$$

*Proof.* By Lemma B.6, each $q_t \in \widetilde{\mathsf{F}}_{\mathrm{disc}}(\theta)$ is supported on a finite set in $\mathbb{R}^{L \times d}$. By Lemma B.7, there exist (indeed, generically) $q_t \in \mathsf{F}_{\mathrm{cont}}(\theta)$ that are absolutely continuous and thus non-atomic. No absolutely continuous distribution can be a finite atomic measure; hence $\widetilde{\mathsf{F}}_{\mathrm{disc}}(\theta) \subsetneq \mathsf{F}_{\mathrm{cont}}(\theta)$ as sets of possible marginals at time $t$. The strictness holds for any $t > 0$ with $g_t > 0$ on a set of positive measure before $t$. □

However, notice that the actual gap may be small, and discrete models with sufficient capacities (such as LLMs) can still approximate distributions pretty well given sufficient vocabulary size, sequence length and training data. Thus in additional to the (marginal) expressivity gain, utilization of continuous representations is also a practical consideration for improving empirical performance: the well-pretrained representations could facilitate the diffusion model training via representation alignment and guidance (Li et al., 2024a; Yu et al., 2024; Wang et al., 2025; Kouzelis et al., 2025).

We now give an information-theoretic quantification of information loss in discrete diffusion sampling. We quantify by differential entropy $h$ for continuous logits $l_\theta$ generated by CDM, and by Shannon entropy $H$ for categorical distributions $\mathrm{Cat}(\mathrm{softmax}(l_\theta))$ sampled by DDM.

**Lemma B.9** (Information loss from token sampling in discrete reverse steps). *Let $\ell_\theta(x_t, t) \in \mathbb{R}^V$ denote the logits predicted at a discrete reverse step, and let the next input be the sampled token $x_{t^-} \sim \mathrm{Cat}(\mathrm{softmax}(\ell_\theta))$. Assume $\ell_\theta$ is a continuous random vector with a non-degenerate distribution (e.g., due to data randomness). Then*

$$I(\ell_\theta; x_{t^-}) \leq H(x_{t^-}) \leq \log V < h(\ell_\theta),$$

*hence the mapping $\ell_\theta \mapsto x_{t^-}$ is information-losing, and the full logit geometry is not preserved along the trajectory.*

*Proof.* By data processing inequality for the Markov chain $\ell_\theta \to \mathrm{softmax}(\ell_\theta) \to x_{t^-}$ (followed by categorical sampling), $I(\ell_\theta; x_{t^-}) \leq I(\ell_\theta; \mathrm{softmax}(\ell_\theta)) \leq H(x_{t^-})$. Since $x_{t^-}$ takes values in a finite set of size $V$, $H(x_{t^-}) \leq \log V$. Meanwhile $\ell_\theta$ is continuous/non-degenerate, so its (differential) entropy $h(\ell_\theta)$ can be arbitrarily large, and the discrete entropy $H(\lfloor \ell_\theta \rfloor)$ is also unbounded with quantization fineness; in particular, $h(\ell_\theta)$ is not bounded by $\log V$. Therefore $I(\ell_\theta; x_{t^-}) < h(\ell_\theta)$; the mapping is many-to-one and loses information about $\ell_\theta$ beyond what is encoded in the sampled index. □

### B.1.2. CONTINUOUS DIFFUSION GENERALIZES LOOPED TRANSFORMER

**Assumption B.10** (Mild regularity for numerical integration). Assume the reverse PF–ODE (4) uses a vector field $v_\theta(z, t) := a_t(z) - \frac{1}{2} g_t^2 s_\theta(z, t)$ that is globally Lipschitz in $z$ and piecewise continuous in $t$. Let $\{t_k\}_{k=0}^T$ be a partition of $[0, 1]$ with $\Delta t_k = t_{k+1} - t_k$ and a standard one-step method $\Psi_{\Delta t_k}(z, t_k)$ (e.g., explicit Euler) that is consistent of order $\geq 1$.

**Proposition B.11** (Continuous diffusion sampler can *simulate* looped rollouts, Proposition 3.4 in main text). *Fix any looped transformer $\Phi_\theta$ and any $T \in \mathbb{N}$. Define the (deterministic) sampler for the reverse PF–ODE by the explicit Euler method with step size $1/T$ and choose the vector field on grid points by*

$$v_\theta(z, t_k) := \Phi_\theta(z) - z, \qquad k = 0, \dots, T - 1.$$

*Then the sampler update is*

$$z_{t_{k+1}} = z_{t_k} + \Delta t\, v_\theta(z_{t_k}, t_k) = \Phi_\theta(z_{t_k}),$$

*which exactly reproduces the looped rollout $h_{k+1} = \Phi_\theta(h_k)$ when we identify $h_k = z_{t_k}$.*

*Proof.* By construction, with $\Delta t = 1/T$ and the explicit Euler method,

$$z_{t_{k+1}} = z_{t_k} + \Delta t\, v_\theta(z_{t_k}, t_k) = z_{t_k} + \tfrac{1}{T}\big(\Phi_\theta(z_{t_k}) - z_{t_k}\big).$$

If we instead scale the vector field as $v_\theta^{(\text{scaled})}(z, t_k) := T\big(\Phi_\theta(z) - z\big)$ while keeping $\Delta t = 1/T$, then

$$z_{t_{k+1}} = z_{t_k} + \Delta t\, v_\theta^{(\text{scaled})}(z_{t_k}, t_k) = z_{t_k} + \tfrac{1}{T} \cdot T\big(\Phi_\theta(z_{t_k}) - z_{t_k}\big) = \Phi_\theta(z_{t_k}).$$

Thus each sampler step equals one looped-transformer application. Since the construction uses the same network $\theta$ inside $\Phi_\theta$ (embedded into $v_\theta$ through the formula above) and a time index via $t_k$, the equality holds step by step. $\square$

**Proposition B.12** (Looped transformer can emulate diffusion ODE terminal maps with timestep embeddings and residual connections). *Let the reverse sampler integrate the PF–ODE (4) with a one-step method $\Psi_{\Delta t_k}$ under Assumption B.10. Define a looped transformer $\Phi_\theta^{\text{ode}}(\cdot; k)$ that, at step $k$, applies the numerical increment*

$$\Phi_\theta^{\text{ode}}(z; k) := \Psi_{\Delta t_k}(z, t_k) = z + \Delta t_k\, v_\theta(z, t_k) + \mathcal{O}(\Delta t_k^2),$$

*where $v_\theta(\cdot, t_k)$ is computed by the same $f_\theta(\cdot, t_k)$ (time-conditioned). Then unrolling $T$ steps computes the same discrete trajectory as the ODE sampler up to the integrator's local truncation error; as $T \to \infty$ (mesh size $\max_k \Delta t_k \to 0$), the terminal error vanishes by standard numerical ODE theory.*

*Proof.* At each step $k$, the looped transformer block applies the map $z \mapsto \Psi_{\Delta t_k}(z, t_k)$ using $f_\theta(\cdot, t_k)$ to evaluate $v_\theta$. Hence

$$h_{k+1} = \Phi_\theta^{\text{ode}}(h_k; k) = \Psi_{\Delta t_k}(h_k, t_k).$$

This matches the sampler's numerical update. The global error after $T$ steps is bounded by $C \max_k \Delta t_k$ for a Lipschitz vector field (by Grönwall-type stability bounds and order-1 consistency). Taking the mesh to zero drives the terminal error to zero. $\square$

*Remark B.13* (Stochastic paths and determinism). If $g_t > 0$, the reverse (3) yields a *distribution over trajectories*. A purely deterministic looped rollout $h_{k+1} = \Phi_\theta(h_k)$, given fixed initial $h_0$, cannot match a non-degenerate stochastic path law. If one augments the looped transformer with exogenous randomness (e.g., $u \sim \mathcal{N}(0, I)$ at initialization or fresh per-step noise) and allows conditioning on $u$ at each step, terminal distributions can be matched in principle by pushing $u$ through the unrolled network.

**Theorem B.14** (Strictness vs. parity: diffusion vs. looped transformer). *Under the* same *parameter budget and the "single time-conditioned network" protocol:*

(i) *(Deterministic ODE samplers) If the reverse uses the PF–ODE ((4) with $g_t \equiv 0$) and is implemented by a standard one-step method, then continuous diffusion is* not *strictly more expressive in terms of terminal distributions: by Propositions 3.4 and B.12, each can simulate the other's discrete rollout (up to vanishing numerical error).*

*(ii)* (*Stochastic path laws*) *If $g_t > 0$ and the looped transformer is deterministic (no exogenous noise), continuous diffusion is strictly more expressive at the trajectory level (cannot match the non-degenerate stochastic path law with a deterministic map).*

*Proof.* (i) Proposition 3.4 shows a diffusion ODE sampler with Euler steps can exactly recover a looped rollout (by choosing $v_\theta^{(\text{scaled})}(z, t_k) = T(\Phi_\theta(z) - z)$). Conversely, Proposition B.12 shows a looped transformer can emulate the numerical ODE integrator step map $\Psi_{\Delta t_k}$; standard numerical analysis ensures convergence of the terminal state as the mesh refines.

(ii) Suppose the diffusion reverse is an SDE with $g_t > 0$ on a set of positive measure. Then the terminal random variable $Z_0$ has a non-degenerate conditional distribution given $Z_1$ (and the Brownian path). A deterministic looped rollout $h_{k+1} = \Phi_\theta(h_k)$ with fixed $h_0$ is a measurable function of $h_0$ only and thus produces a Dirac path law; it cannot match a non-degenerate distribution over trajectories. Hence diffusion is strictly more expressive pathwise. $\qquad\square$

Notably, if the looped model is allowed an auxiliary noise input $u$ (initial or per-step) and time conditioning, then the unrolled map can be written as $H_T = \Gamma_\theta(u)$ for some measurable $\Gamma_\theta$. For any target terminal law with a Borel probability measure on $\mathbb{R}^{L \times d}$, there exists a pushforward of a simple noise (e.g., Gaussian) that realizes it; with sufficient capacity, the looped model can approximate such a map. Thus terminal parity is achievable in principle.

*Remark* B.15 (Timestep embeddings and intermediate supervision). As shown in (Xu & Sato, 2024), timestep embeddings provably improve the expressiveness of looped transformers, though the gain might be marginal in practice. Supervising intermediate diffusion steps (multi-$t$ losses) does not change the representable set either; it improves optimization and inductive bias. Stochastic SDE sampling ($g_t > 0$) enlarges the *path* distribution class (Remark B.13) but does not imply a strict advantage on terminal laws once exogenous noise is also allowed in looped rollouts.

We now further discuss the superiority of CCDD over looped transformer in terms of both performance and efficiency. As shown in Table 6, while both CCDD and LT achieves perfect performance on Sudoku, CCDD is better on 3-SAT and Countdown. Moreover, another even larger advantage of CCDD lies in its efficient supervision and scalability/flexibility. In particular, a $T$-depth looped transformer training requires to roll our the entire depth, bringing $T\times$ memory and time consumption (and even so cannot provide intermediate supervision), while a CCDD with $T$ diffusion steps can fully simulate the former yet only requiring $1\times$ training cost, thanks to the efficient training formulation of diffusion models: in every iteration, we only need to sample one intermediate timestep, and the single forward pass can provide intermediate supervision. As a result, it is impossible to pretrain a looped transformer with effective depth 1000 as we do in CCDD on OWT with 32 H100 GPUs, even for the small size models. Therefore, CCDD fully covers looped transformer in performance, while being scalable to real tasks. Moreover, loop transformer must predefine the looped depth, while CCDD is flexible during both pretraining and inference using the continuous-time Markov Chain (CTMC) and continuous-time SDE/ODE framework.

We conclude this subsection (corresponding to Section 3.1 in the main text) by summarizing the comparison in expressivity as follows.

- **Discrete vs. Continuous Diffusion:** By Theorem 3.2, continuous diffusion strictly dominates discrete diffusion at the *trajectory* level in $\mathbb{R}^{L \times d}$ (non-atomic intermediate marginals vs. finite-atomic). Lemma B.9 shows discrete per-step token sampling loses full logit information, whereas continuous samplers preserve all coordinates without mandatory quantization.

- **Continuous Diffusion vs. Looped Transformer:** Continuous diffusion samplers can *simulate* looped rollouts (Proposition 3.4); looped rollouts can *emulate* ODE samplers (Proposition B.12). Diffusion with $g_t > 0$ is strictly more expressive *pathwise* than deterministic looped rollouts (Theorem B.14(ii)).

## B.2. Analysis of CCDD

This subsection corresponds to Section 4 in the main text, providing theoretical results and more analysis of CCDD.

### B.2.1. EXPRESSIVITY OF CCDD

**Expressivity comparison.** We now compare the joint model against (i) *continuous diffusion alone* (on $\mathcal{Z}$) and (ii) choices of reverse parameterization.

**Theorem B.16** (Expressivity vs. continuous-only diffusion: joint law and marginals). *Consider the hypothesis classes of terminal laws* $\mathcal{H}_{\text{joint}} := \{$ *laws of* $(x_0, z_0)$ *produced by the joint model* $\}$ *and* $\mathcal{H}_{\text{cont}} :=$ $\{$ *laws of* $z_0$ *produced by continuous diffusion alone* $\}$. *Then:*

(i) *On the* joint space $\mathcal{X} \times \mathcal{Z}$, $\mathcal{H}_{\text{joint}}$ *strictly extends continuous-only modeling (which does not produce* $x_0$ *at all); i.e., the joint model is strictly more expressive if the target includes the discrete marginal.*

(ii) *On the* continuous marginal *alone, the joint model does* not *enlarge the class of* $z_0$ *laws beyond a continuous diffusion with the same parameter budget and time-conditioning: for any* $P \in \mathcal{H}_{\text{joint}}$, *its* $z$-marginal $P_Z$ *lies in* $\mathcal{H}_{\text{cont}}$ *(up to decoder equivalence).*

*Proof.* (i) Trivial: continuous-only models do not define a distribution on $x_0$; the joint model does.

(ii) Let $P \in \mathcal{H}_{\text{joint}}$ be realized by some reverse parameterization with $f_\theta(x_t, z_t, t)$. Define a continuous-only model with inputs $(z_t, t)$ whose network computes the same $z$-head as the joint model but with $x_t$ embedded by a deterministic encoder $\mathcal{E}$ predicted from $(z_t, t)$ (this is implementable because the joint reverse factor $p_\theta^{\text{cont}}$ depends on $(x_t, z_t, t)$ only through a measurable function). Universal approximation guarantees allow the continuous-only network to approximate the composite map $(z_t, t) \mapsto p_\theta^{\text{cont}}(\cdot \mid x_t, z_t, t)$ after marginalizing out $x_t$ under $q_t(x_t \mid z_t)$ (which the network can input as $\mathbb{E}[\phi(x_t) \mid z_t, t]$ for a rich feature dictionary $\phi$). Therefore the induced terminal $z$-law can be matched. The decoder equivalence argument is standard: terminal classification/regression heads can implement the same marginal law. $\square$

*Remark* B.17. Theorem B.16(ii) states a *marginal* parity: adding a discrete companion does not expand the set of achievable $z$-marginals at the level of function classes (with time-conditioning and equal total parameters). Practically it may ease optimization by providing an auxiliary target.

**Semigroup structure and sufficient expressivity of factored parameterization.** Let $q_t(x, z)$ denote the joint forward marginal induced by (13). Then

$$\nabla_z \log q_t(x, z) = \nabla_z \log q_t(z \mid x), \qquad \Delta_z \log q_t(x, z) = \Delta_z \log q_t(z \mid x), \tag{19}$$

i.e., the continuous score depends on $x$ *only through* the conditional $q_t(z \mid x)$. For the discrete part, Bayes posteriors use $q_t(x \mid z) \propto q_t(x) \, q_t(z \mid x)$.

**Lemma B.18** (Forward semigroup and Trotter factorization). *Let* $\{T_t^{(z)}\}_{t \geq 0}$ *and* $\{T_t^{(x)}\}_{t \geq 0}$ *be the Markov semigroups generated by the continuous FP operator* $L_t^{(z)}$ *and the discrete generator* $L_t^{(x)}$ *(Kolmogorov forward) respectively, both time-inhomogeneous but piecewise constant in small intervals. Then the joint forward semigroup on* $\mathcal{X} \times \mathcal{Z}$ *with* independent *corruption is* $T_t = T_t^{(x)} T_t^{(z)} = T_t^{(z)} T_t^{(x)}$ *and, for a partition* $0 = t_0 < \cdots < t_N = t$ *with mesh* $\max_k \Delta t_k \to 0$,

$$T_t = \lim_{\max \Delta t_k \to 0} \prod_{k=0}^{N-1} \left( T_{\Delta t_k}^{(x)} T_{\Delta t_k}^{(z)} \right) = \lim_{\max \Delta t_k \to 0} \prod_{k=0}^{N-1} \left( T_{\Delta t_k}^{(z)} T_{\Delta t_k}^{(x)} \right).$$

*Proof.* Independence in (13) implies that the joint generator is the *sum* $L_t = L_t^{(z)} + L_t^{(x)}$ acting on functions $f(x, z)$ by $(L_t f)(x, z) = (L_t^{(z)} f)(x, z) + (L_t^{(x)} f)(x, z)$. For (piecewise) time-constant generators the Chernoff–Lie–Trotter product formula yields the stated limits; commutativity at the semigroup level follows from independence. $\square$

**Theorem B.19** (Effect of factored reverse kernels). *Fix a time step* $t \to s$ *and consider the family of joint reverse kernels* $\mathcal{K} = \{ K(x_s, z_s \mid x_t, z_t) \}$. *Let*

$$\mathcal{K}_{\text{fact}} := \left\{ K(x_s, z_s \mid x_t, z_t) = K_x(x_s \mid x_t, z_t) \, K_z(z_s \mid x_t, z_t) \right\}$$

*be the* factored *family in (15). Then:*

(a) *Single-step limitation.* $\mathcal{K}_{\text{fact}}$ *is a strict subset of* $\mathcal{K}$*: there exist joint kernels with within-step couplings that cannot be written as a product independent of* $(x_s, z_s)$ *cross-dependence.*

(b) Splitting sufficiency (small steps). *Suppose the target joint dynamics has generator $L_t = L_t^{(z)} + L_t^{(x)}$ (no cross-diffusion term), and the factored reverse uses* conditionally coupled *factors $K_x(\cdot \mid x_t, z_t)$ and $K_z(\cdot \mid x_t, z_t)$. Then, by Lie–Trotter splitting, iterating factored kernels at a small step size $\Delta t$ alternately (e.g., $K_z K_x$ per micro-step) converges to the* same *joint semigroup as any coupled kernel generated by $L_t$, hence there is no expressivity loss at the trajectory level as $\Delta t \to 0$.*

*Proof.* (a) Consider $\mathcal{X} = \{0, 1\}$ and $z \in \mathbb{R}$. Define a joint kernel that enforces $x_s = \mathbf{1}\{z_s > 0\}$ almost surely (hard constraint). This coupling cannot be expressed as $K_x(x_s \mid x_t, z_t) K_z(z_s \mid x_t, z_t)$ because any factorization leaves $x_s$ independent of $z_s$ *given* $(x_t, z_t)$, contradicting the deterministic relation between $x_s$ and $z_s$.

(b) Under the assumed generator sum $L_t^{(z)} + L_t^{(x)}$, the exact joint semigroup over a small interval $\Delta t$ equals $e^{\Delta t (L_t^{(z)} + L_t^{(x)})}$, while the alternating product $e^{\Delta t L_t^{(z)}} e^{\Delta t L_t^{(x)}}$ (or the corresponding Markov kernels $K_z K_x$) approximates it with first-order error $\mathcal{O}(\Delta t^2)$. Over a partition with mesh $\max \Delta t \to 0$, the product converges to the exact semigroup (Chernoff–Lie–Trotter), hence the factored parameterization with alternating micro-steps is sufficient to realize the same family of joint laws. $\square$

**Corollary B.20** (Coupling via conditioning is enough in the small-step limit). *Even though $K(x_s, z_s \mid x_t, z_t)$ does not condition on $(x_s, z_s)$ jointly in one shot, allowing each factor to depend on* both *$(x_t, z_t)$ and alternating the updates yields asymptotically the same expressivity as a fully coupled kernel driven by a sum-generator $L_t^{(z)} + L_t^{(x)}$. However, the diffusion process design of our language modeling task is flexible as long as it preserve the clean marginals, thus a factored process suffices.*

*Remark* B.21 (When factorization truly hurts). If the target joint generator includes *cross terms* that cannot be written as $L^{(z)} + L^{(x)}$ (e.g., a diffusion on $\mathbb{R}^{L \times d}$ whose drift or diffusion matrix depends on *future* $x_s$ rather than $x_t$, or a discrete jump rate at time $t$ depending on $z_s$), then any per-step factorization that does not see $(x_s, z_s)$ jointly will *not* reproduce such dynamics without further inner iterations (e.g., Gibbs-within-step to sample $z_s$ and then $x_s$ conditioning on $z_s$).

However, we can always construct a proper diffusion process without cross-terms as in the main text (as long as it preserve the terminal marginals), hence the factored parameterization is expressive enough for our language generation task.

### B.2.2. Designs of Schedules

We now provide more discussions on the design space of noise schedules in CCDD. In addition to approximately synchronous signal-noise ratios in two modalities by adjusting the coefficients $\alpha_t$ and $\eta_t$, we propose asynchronous noise schedules of the forward process in light of the "representation guidance" spirit. We set the information decay rate in continuous space slower than the discrete modality, so that in the reverse process the model would generate the latent representations faster (playing the role of implicit planning and reasoning), which serve as the high-level guidance for token decoding.

**Information theory perspective.** We start with the tools in information theory.

**Lemma B.22** (Mutual information decay under factored corruption). *Let $I_t := I((x_0, z_0); (x_t, z_t))$ denote the mutual information between data and the corrupted pair at time $t$. Under* (13) *with independent noises, $I_t$ is non-increasing and satisfies*

$$\frac{d}{dt} I_t = \mathbb{E}\left[ \frac{d}{dt} \log \frac{q_t(x_t, z_t \mid x_0, z_0)}{q_t(x_t, z_t)} \right] \leq 0,$$

*with decomposition*

$$\frac{d}{dt} I_t = \frac{d}{dt} I((x_0, z_0); z_t \mid x_t) + \frac{d}{dt} I((x_0, z_0); x_t \mid z_t).$$

*In particular, for the VP SDE the continuous contribution equals the negative* Fisher score power*:*

$$\frac{d}{dt} I((x_0, z_0); z_t \mid x_t) = -\mathbb{E}\left[ \|g_t \nabla_z \log q_t(z_t \mid x_t)\|^2 \right],$$

*and for the CTMC the discrete contribution equals (minus) a $\chi^2$-divergence–based entropy production rate.*

*Proof.* Data processing along a Markov chain $(x_0, z_0) \to (x_t, z_t)$ yields non-increasing mutual information. The derivative formula follows from differentiating the KL defining mutual information and the Kolmogorov forward equations; for diffusions the de Bruijn–Fisher identity gives the continuous term; for CTMCs one uses the standard entropy production formula $\frac{d}{dt} H(p_t) = -\sum_{i \neq j} p_t(j)(G_t)_{ij} \log \frac{p_t(i)}{p_t(j)}$ and the $\chi^2$ representation of conditional MI rates. $\square$

**"Optimal" coupling via Hyperparameter Matching.**    It is natural to have the question: is there an optimal way to couple the two modalities? We discuss the choice of forward hyperparameters (diffusion rate $\beta_t$ for $z$, jump rate $u_t$ for $x$) and their coupling.

**Definition B.23** (Signal-to-noise and posterior conditioning). Define the continuous SNR at time $t$ by $\mathrm{SNR}_z(t) := \alpha_t^2/\sigma_t^2$ (VP case), and the discrete SNR by the interpolation weight $\eta_t$ in $q_t(x_t \mid x_0) = \mathrm{Cat}(\eta_t x_0 + (1 - \eta_t)\pi_t)$ (USDM/masked). Define the *joint conditioning strength*

$$\kappa_t := I\big((x_0, z_0);\ z_t \mid x_t\big)\ +\ I\big((x_0, z_0);\ x_t \mid z_t\big),$$

which quantifies how informative each modality remains about its clean counterpart when conditioning on the other.

**Proposition B.24** (Entropy/MI matching heuristic). *(Informal) Let $\beta_t$ and $u_t$ be chosen so that the* rates *of MI decay from Lemma B.22 are balanced:*

$$\mathbb{E}\big[\|g_t \nabla_z \log q_t(z_t \mid x_t)\|^2\big]\ \approx\ \mathrm{Rate}_x(t),$$

*where $\mathrm{Rate}_x(t)$ denotes the CTMC's conditional MI decay rate (a $\chi^2$-type quantity). Then the two posteriors $q(x_0 \mid x_t, z_t)$ and $q(z_0 \mid x_t, z_t)$ maintain comparable conditioning difficulty (well-conditioned Bayes factors), which in turn stabilizes training losses for heads in the discrete modality.*

Intuitively, training both heads amounts to estimating $q(z_0 \mid z_t, x_t)$ and $q(x_0 \mid x_t, z_t)$. If one modality loses information much faster (e.g., $\mathrm{SNR}_z \ll \mathrm{SNR}_x$), then one posterior becomes broad/ill-conditioned relative to the other, causing gradient scale mismatch. Balancing the decay rates equalizes expected Fisher information in the continuous head and the discrete information production, yielding comparable curvature of the objectives (via de Bruijn/Fisher for $z$ and entropy production for the CTMC). This is a standard preconditioning argument based on matching Fisher blocks across modalities.

**Corollary B.25** (Schedule matching guideline). *A practical rule is to pick $\beta_t$ and $u_t$ such that the* monotone *functions $t \mapsto \mathrm{SNR}_z(t)$ and $t \mapsto \eta_t/(1 - \eta_t)$ have similar slopes on a log-scale, e.g.,*

$$\frac{d}{dt} \log \mathrm{SNR}_z(t)\ \approx\ \frac{d}{dt} \log \frac{\eta_t}{1 - \eta_t}.$$

*This equates the relative shrinkage of posterior variances (continuous) and odds (discrete), approximately stabilizing Bayes updates in $x_t$.*

**Practical schedule designs.**    From a practical perspective, we could regard the continuous representations as the guidance for the discrete part, the continuous schedule then should be ahead of the discrete schedule (i.e., generated earlier in the reverse process) as useful high-level guidance. This intuition is also validated by other work involving multiple modalities (Geffner et al., 2025). We adopt this strategy in most experiments, featuring linear schedule for discrete part ($\eta_t = 1 - t$) and concave schedule for continuous part ($\alpha_t = \sqrt{1 - t}$).

We conclude this subsection by summarizing takeaway messages on CCDD.

- **Formulation.** The joint model formalizes a mixed SDE–CTMC system with independent forward corruptions and a reverse denoiser that *conditions* on both $(x_t, z_t)$ but factors the per-step kernel across modalities.

- **Expressivity vs. continuous-only.** Joint modeling is strictly more expressive on the *joint space*; on the *z-marginal*, it does not enlarge the class beyond continuous diffusion with comparable capacity (Theorem B.16).

- **Effect of factorization.** A single-step factored kernel cannot represent arbitrary within-step couplings (Theorem B.19(a)), but alternating factored updates at small step sizes attains the same semigroup when the target generator splits (Theorem B.19(b), Corollary B.20).

- **Coupling/schedules.** Matching information decay across modalities (Proposition B.24, Corollary B.25) yields well-conditioned posteriors.

- **Information-theoretic lens.** The continuous score is $\nabla_z \log q_t(z \mid x)$; MI decays additively across modalities under independence (Lemma B.22). Balancing decay rates aligns Fisher/entropy production and stabilizes learning.

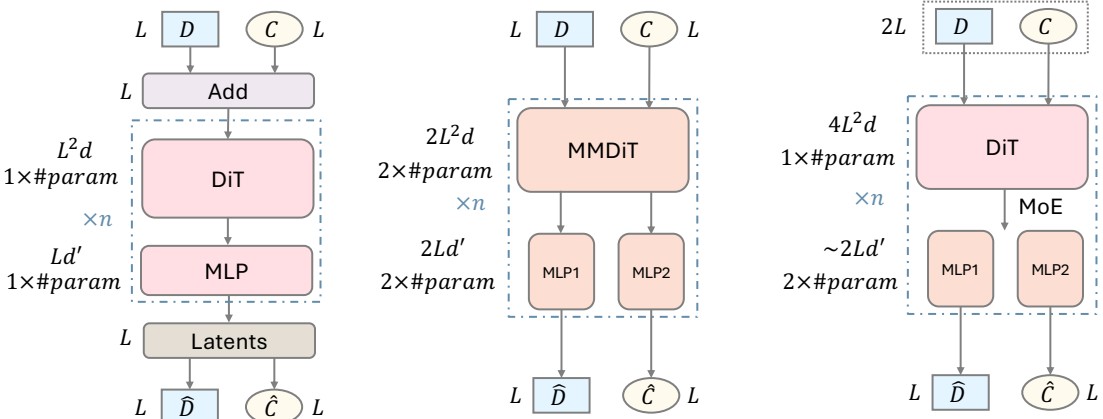

*Figure 5.* Comparison of different denoising network architectures for CCDD: MDiT (left), MMDiT (middle), and MoEDiT (right).

## C. Implementation Details

### C.1. Architecture Design for CCDD

The architecture consists of basic diffusion transformer blocks. Optional timesteps conditioning is embedded through adaLN. For standard DiT blocks, attention is followed by MLPs.

**DiT.** Standard DiT except that we mix continuous and discrete embeddings before the first DiT block (through adding or concatenating), and decode both continuous and discrete tokens from the output of the last DiT block. Therefore, the actual processed tokens are of shape $[B, L, d]$ and the attention complexity is $\mathcal{O}(L^2)$.

**MMDiT.** Since MMDiT is naturally capable of processing multimodal generation, we opt it to generate continuous and discrete tokens simultaneously. We also adopt a slightly different version that stagger cross-attention blocks for modality interaction and self-attention blocks for single modalities, respectively. The tokens are of shape $[B, 2L, d]$ consequently, with the attention complexity $\mathcal{O}(2L^2)$ and parameters doubled.

**DiT with MoE.** To maximize the expressivity while keeping the number of parameters not doubled, we choose to maintain representation for both continuous and discrete tokens, resulting $[B, 2L, d]$ representations. However, two modalities share the same attention parameters, and we compute self-attention over the $2L$ tokens, resulting $\mathcal{O}(4L^2)$ attention complexity without doubling the number of parameters. To avoid modality collapse, we use different MLPs for discrete and continuous modality. However, instead of hard separating MLPs, we use MoE architecture and let each token selects the proportion of experts, keeping maximum expressivity for every token while allowing discrete and continuous tokens to be processed differently according to their states. (For example, if the discrete state is cleaner, then there might be a higher weight on the discrete MLPs, and vice versa.) We call this architecture "MoEDiT".

In our implementation, we may interpolate the cross-modal MMDiT blocks with unimodal self-attention blocks and use their combinations. Our experiments aim to *investigate the performance with same parameter or computation conjectures*, or *studies the scaling behavior w.r.t. parameters and FLOPs*.

Remarkably, when adopting MDiT, CCDD has actually almost the same number of parameters as well as the same FLOPs compared with discrete diffusion baselines with DiT architectures – the only difference is the additional linear input and output heads for the continuous component, whose FLOPs are ignorable compared with main DiT blocks. Thus, both the training time and inference time of CCDD with MDiT architecture almost does not increase, yet the performance gain compared with discrete baselines is nontrivial. MMDiT has doubled parameters and $2\times$ FLOPs in core transformer blocks, while MoEDiT has the same attention parameters and doubled FFN parameters, with $4\times$ FLOPs in attention and approximately $2\times$ FLOPs in FFN.

As shown in Table 2 and Table 3, CCDD with MDiT is able to significantly surpass baselines without increasing non-embedding parameters and FLOPs; despite more parameters and FLOPs, MMDiT and MoeDiT both showcase much stronger performance compared with discrete baselines. In comparison, both MMDiT and MoEDiT bring prominent performance gain, among which MMDiT balances efficiency and performance, and MoEDiT balances parameters and performance.

## C.2. Algorithm

We now give the algorithmic description of CCDD training in Algorithm 1 and sampling in Algorithm 2 using DDPM / DDIM for the continuous example. Notably, the main computation during inference occurs in the denoising model forward, i.e., line 3 in Algorithm 2, thus the inference cost also depends on the architecture design explained in Section C.1. With proper architecture selection, both *CCDD training and inference could be extremely efficient*.

---

**Algorithm 1** CCDD Training

---

**Require:** Dataset $\mathcal{D}$ of pairs $(x_0, z_0)$ with $z_0 = \mathcal{E}(x_0)$; time sampler $t \sim \mathcal{U}([0, 1])$ or discrete grid $\{t_k\}$; VP schedule $\{\alpha_t, \sigma_t\}$; discrete schedule $\{\eta_t\}$ and noise distribution $\{\pi_t\}$; loss weights $\lambda_{\text{cont}}(t), \lambda_{\text{disc}}(t)$.

1: Initialize parameters $\theta$ of a single time-conditioned model $f_\theta(\cdot, t)$ with two heads:

$$\epsilon_\theta(x_t, z_t, t) \in \mathbb{R}^{L \times d}, \qquad \ell_\theta(x_t, z_t, t) \in \mathbb{R}^{L \times V}.$$

2: **for** each minibatch $\{(x_0^{(b)}, z_0^{(b)})\}_{b=1}^B \subset \mathcal{D}$ **do**

3:     Sample times $\{t^{(b)}\}_{b=1}^B$.

4:     **Forward corruption:** Sample $\epsilon^{(b)} \sim \mathcal{N}(0, I)$ and set

$$z_t^{(b)} \leftarrow \alpha_{t^{(b)}} z_0^{(b)} + \sigma_{t^{(b)}} \epsilon^{(b)}, \quad x_t^{(b)} \sim \text{Cat}\Big(\eta_{t^{(b)}} e_{x_0^{(b)}} + (1 - \eta_{t^{(b)}}) \pi_{t^{(b)}}\Big).$$

5:     **Model prediction:**

$$\epsilon_\theta^{(b)} \leftarrow \epsilon_\theta\big(x_t^{(b)}, z_t^{(b)}, t^{(b)}\big), \qquad \ell_\theta^{(b)} \leftarrow \ell_\theta\big(x_t^{(b)}, z_t^{(b)}, t^{(b)}\big).$$

6:     **Continuous loss (VP, $\epsilon$-pred MSE) and discrete loss (token CE on masked tokens):**

$$\mathcal{L}_{\text{cont}} \leftarrow \frac{1}{B} \sum_{b=1}^B \lambda_{\text{cont}}(t^{(b)}) \big\| \epsilon^{(b)} - \epsilon_\theta^{(b)} \big\|_2^2.$$

$$\mathcal{L}_{\text{disc}} \leftarrow -\frac{1}{B} \sum_{b=1}^B \lambda_{\text{disc}}(t^{(b)}, x_0^{(b)}, x_t^{(b)}) \log \text{softmax}\big(\ell_\theta^{(b)}\big)[x_0^{(b)}].$$

7:     **Total loss and update:** $\mathcal{L} \leftarrow \gamma_{\text{cont}} \cdot \mathcal{L}_{\text{cont}} + \gamma_{\text{disc}} \cdot \mathcal{L}_{\text{disc}}$

8:     Update $\theta \leftarrow \theta - \eta \nabla_\theta \mathcal{L}$.

9: **end for**

---

---

**Algorithm 2** CCDD Sampling

---

**Require:** Time grid $1 = t_0 > t_1 > \cdots > t_K = 0$; forward schedules $(\alpha_{t_k}, \sigma_{t_k})$, $(\eta_{t_k}, \pi_{t_k})$; reverse kernels $q_{t_k|t_{k+1}}$ for discrete; DDPM/DDIM choice for continuous with variance knob $\eta_{\text{ddpm}} \in [0, 1]$.

1: **Init:** $x_{t_0} \sim \pi_{t_0}$ (uniform or [MASK] prior per token); $z_{t_0} \sim \mathcal{N}(0, I)$.
2: **for** $k = 0$ **to** $K - 1$ **do**
3:     **Model heads:**
$$\epsilon_\theta \leftarrow \epsilon_\theta(x_{t_k}, z_{t_k}, t_k), \qquad \ell_\theta \leftarrow \ell_\theta(x_{t_k}, z_{t_k}, t_k), \quad \hat{\pi}_\theta = \text{softmax}(\ell_\theta).$$
4:     **(A) Discrete reverse (Bayes form):**
$$p_\theta(x_{t_{k+1}} \mid x_{t_k}, z_{t_k}) \propto q_{t_k|t_{k+1}}(x_{t_k} \mid x_{t_{k+1}}) \, \hat{\pi}_\theta(x_{t_{k+1}} \mid x_{t_k}, z_{t_k}, t_k).$$
5:     Sample (or take mode) $x_{t_{k+1}} \sim p_\theta(\cdot \mid x_{t_k}, z_{t_k})$.
6:     **(B) Continuous reverse (VP, $\epsilon$-pred):**
$$\hat{z}_{0,\theta} \leftarrow \frac{z_{t_k} - \sigma_{t_k} \epsilon_\theta}{\alpha_{t_k}}.$$
7:     **DDIM mean:** $m_{t_{k+1}} \leftarrow \alpha_{t_{k+1}} \hat{z}_{0,\theta} + \sigma_{t_{k+1}} \epsilon_\theta$.
8:     **Stochastic DDPM step (optional):** $z_{t_{k+1}} \leftarrow m_{t_{k+1}} + \eta_{\text{ddpm}} \sigma_{t_k|t_{k+1}} \xi, \; \xi \sim \mathcal{N}(0, I)$.
9:     **DDIM step (deterministic):** set $\eta_{\text{ddpm}} = 0$ to use $z_{t_{k+1}} \leftarrow m_{t_{k+1}}$.
10: **end for**
11: **Decode:** Return tokens $\hat{x}_0 \leftarrow x_{t_K}$ and/or logits from a decoder applied to $z_{t_K}$ (if needed).

---

# D. Experimental Details

We provide more experimental details and additional results in this section.

## D.1. Experimental Configurations

**Schedules.** For the discrete process, we use masked noise by default with $\gamma_t = 1 - t$ as in most discrete diffusion papers. For the continuous process, we use VP procedure as in DDPM (Ho et al., 2020) or the (log-)linear schedule in most flow matching papers. Ablation study on the continuous schedule are reported in Table 7.

**Embedding spaces.** Since Qwen3-Embedding enables flexible output dimensions down to 32, we use the 32-dimensional last-layer embeddings without specification. This selection is consistent with the analysis in the main text. Low-dimensional latent space is the standard setting in recent vision diffusion models (Esser et al., 2024). For RoBERTa-base embeddings, we use the full embedding with hidden size 768. All representations are normalized.

**Other hyper-parameters.** We set $p_{\text{drop}} = 0.15$ as in the masked rate in BERT (Devlin et al., 2019) training. Without specification, we set the loss weights $\gamma_{\text{cont}} = \gamma_{\text{disc}} = 1$ and use gradient clipping. Following (Sahoo et al., 2024; von Rütte et al., 2025), on LM1B we set a constant learning rate $3 \times 10^{-4}$ with 2500 warm-up steps, and a constant learning rate $5 \times 10^{-4}$ with 10000 warm-up steps for OWT. We use AdamW optimizers with weight decay 0.02 and gradient norm 1.0.

**Computation resources.** All pretraining tasks are conducted with 8 NVIDIA H100 or A100 80GB GPUs. As an example, pretraining on OWT with 8 H100 GPUs requires 135 hours for MDiT, 255 hours for MMDiT, and 226 hours for MoEDiT.

## D.2. Discussion on Validation Perplexity

Recall the forward process of CCDD. Since we use masking process for the discrete component with $\gamma_t = 1 - t$, the corresponding discrete loss is calculated as the mean of cross-entropy loss of the masked tokens:

$$\mathcal{L}_{\text{disc}} = \mathbb{E}_{t, x_t} \left[ \mathbf{1}_{x_t = m} \cdot \boldsymbol{x}^\top \log \boldsymbol{x}_\theta \right] \tag{20}$$

When calculating validation elbo, the model needs to predict the masked tokens. However, the model actually also takes the partially noised continuous tokens as the input, which provide information of the masked tokens that is unavailable in discrete diffusion, causing potential unfairness.

*Table 7.* Validation ELBO of ablations on LM1B. We use Qwen3-Embedding-0.6B as the continuous generation space for CCDD, and train all models for 33B tokens. The numbers non-embedding parameter counts are also reported for fair comparison. "VP" refers to the variance preserving schedule in DDPM, and "Linear" refers to the log-linear schedule in flow matching.

| Base Model | Architecture | # params. | Cont. Schedule | Validation ELBO ($\downarrow$) |
|---|---|---|---|---|
| CCDD | MDiT | 92.1M | VP | 2.338 |
| CCDD | MDiT | 92.1M | Linear | 2.586 |
| CCDD | MMDiT | 216.2M | VP | 2.311 |
| CCDD | MMDiT | 216.2M | Linear | 2.518 |

To address this issue, we use special methods to erase the related information in the continuous tokens corresponding to the masked discrete ones. For RoBERTa, we sample $\tilde{z}_0 = \mathcal{E}(x_t)$ as the embedding of $x_t$ (namely the partially masked sequence) instead of $x_0$, so that the "clean" representations of the masked tokens would not directly be determined by the oracle. For Qwen3-Embedding model which does not have a mask token, we set the embeddings corresponding to masked tokens of discrete component in $z_0 = \mathcal{E}(x_0)$ to zeros in order to simulate the "masking" operation, leading to $\tilde{z}_0$ which also declines the direct information leakage. The continuous forward process then starts with $\tilde{z}_0$ instead of the original $z_0$. To let the model capable of doing inference with these perturbed inputs, we also perform these masking operations with a certain probability $p_r$ during training, which is stochastically sampled per-sequence within $[0, 0.9]$.

Notably, evaluating ELBO with $\tilde{z}_0$ actually makes the inference of CCDD harder than discrete only, since the unmasked tokens in the discrete part are injected noise to their continuous representations, while the purely discrete model only takes the clean inputs. In the main text, we always report results with $p_r = 1$, which are proper. Even if evaluated with a strictly harder metric, CCDD still outperforms the baselines, which effectively validates the superiority of joint modeling.

### D.3. Additional Ablations

We further study the effects of architecture and continuous schedules. We set $p_r = 0$ for all models during training and inference for simplicity. The validation perplexity is calculated as the exponential function of ELBO. Table 7 validates that a continuous schedule ahead of the discrete part in inference time (VP) yields better results. Scaling the number of parameters also consistently improves the performance.

*Table 8.* Validation ELBO with $p_r = 0$ on OWT. We use Qwen3-Embedding-0.6B as the continuous generation space for CCDD, and train all models for 131B tokens.

| Model | Validation ELBO ($\downarrow$) | Validation PPL ($\downarrow$) |
|---|---|---|
| CCDD-MDiT | 2.457 | 11.67 |
| CCDD-MMDiT | 2.415 | 11.19 |

We also report the validation ELBO with $p_r = 0$ (always starting diffusion process on the oracle $x_0$ and $z_0$) in Table 8 for reference. The models are also trained with Qwen3-Embeddings on OWT leveraging VP schedules. We observe that even a simple MDiT (which essentially shares the same non-embedding parameters as standard DiT except the additional encoding layer and the decoding head) could obtain super low ELBO.

### D.4. Additional Results and Details of Downstream Benchmarks and Complex Reasoning Datasets

**Downstream benchmarks.** We evaluate the general performance of our language model for language understanding based on the commonly adopted lm-eval evaluation suite. We follow the benchmark suite choice of (von Rütte et al., 2025), which covers a diverse range of tasks regarding general language understanding and question answering capability and can convincingly showcase the performance comparison among different models. The zero-shot downstream benchmarks include ARC (Clark et al., 2018) (both elementary and challenge subsets), BoolQ (Clark et al., 2019), PIQA (Bisk et al., 2019), OpenBookQA (Mihaylov et al., 2018), and WinoGrande (Sakaguchi et al., 2019). They cover a wide range of question answering, complex mathematical and commonsense reasoning, and domain knowledge. Our evaluation focuses on likelihood-based multiple-choice tasks, where the per-token likelihood is calculated across both the context and the completion, but not the padding.

*Table 9.* Zero-shot benchmark accuracy on downstream datasets.

| Model | # params. | ARC-e | ARC-c | BoolQ | PIQA | OBQA | WinoG | Avg. |
|---|---|---|---|---|---|---|---|---|
| GPT2 | 110M | **43.81** | 19.03 | 48.72 | **62.89** | 16.40 | 51.62 | 40.41 |
| Llama (retrain) | 117M | 40.53 | 25.51 | 46.21 | 62.73 | **28.40** | 50.57 | **42.35** |
| MDLM | 92.1M | 28.37 | 23.63 | 49.42 | 52.50 | 22.00 | 49.41 | 37.55 |
| GIDD | 92.1M | 26.73 | 23.55 | 50.43 | 51.85 | 26.60 | 49.49 | 38.11 |
| GIDD-base | 320M | 28.32 | 23.12 | 49.57 | 52.83 | 23.80 | 48.30 | 37.66 |
| CCDD-MMDiT (ours) | 144.1M | 28.75 | 25.94 | **51.10** | 54.41 | 26.60 | 49.96 | 39.46 |
| CCDD-MoEDiT (ours) | 104.0M | 30.01 | **26.02** | 50.21 | 55.66 | 23.60 | **51.93** | 39.57 |

As shown in Table 9, CCDD (w/ mmdit architecture and Roberta representations) outperforms all discrete diffusion baselines in all six benchmarks, and achieves comparable or better performance compared with AR models. The results rigorously validate the advantages of CCDD over traditional MDM baselines. Notably, this CCDD model is simply trained with Roberta, yet still surpasses all discrete diffusion and some AR models - we do not report CCDD with stronger Qwen3-Embedding again due to incomparable ELBOs caused by different tokenizers.

**Complex reasoning datasets.** We now provide details of the complex reasoning datasets in Table 6. The goal of Sudoku is to fill a $9 \times 9$ grid with numerical digits, ensuring that every column, row, and $3 \times 3$ subgrid contains all the numbers from 1 to 9. The goal of SAT (Boolean satisfiability problem) is to determine whether a given Boolean formula represented in conjunctive normal form (CNF) can be assigned a set of values (0 or 1) to its variables, such that the formula evaluates to true. We select 3-SAT problems with 9 variables (i.e., 3-SAT 9). Countdown is a mathematical reasoning challenge generalized from the game of 24, which even GPT-4 struggles with (Yao et al., 2023). The goal of Countdown is using the given numbers and arithmetic operations $(+ - */)$ to obtain the target number. We select the subproblems with 4 variables (i.e., Countdown-4).

# E. Discussion

## E.1. Strengths of CCDD

We systematically summarize the advantages of CCDD as follows, which may inspire more future extensions.

- Exhibits strong expressivity, retains full information on marginal distribution and rich contextualized semantics.

- Ability to potentially conduct implicit reasoning, searching and planning in the latent space.

- Combines a smooth and conservative decoder containing rich semantics with an aggressive decoder modeling explicit tokens (which decomposes the process to a series of conditional generation).

- Receives knowledge distillation from pretrained models, and representation learning accelerates diffusion model training.

- Flexible in few-step sampling and inference-time scaling while compatible with CFG.

## E.2. Further Discussions on Generation Spaces of CDM

Let $\mathcal{E}_{\text{tok}} : \Omega \to \mathbb{R}^d$ be a *token-wise* embedding with a fixed codebook $C = \{e_1, \ldots, e_V\} \subset \mathbb{R}^d$ (typically $d \leq V$, often $d \ll V$). Let $\mathcal{E}_{\text{ctx}} : \Omega^L \to \mathbb{R}^d$ be a *contextualized* embedding at a given position (depending on the entire sequence context), again with $d \leq V$ in practice. We analyze three generation spaces for the *continuous* diffusion variable at a position:

$$\text{(i) } \mathcal{Z}_\Delta = \Delta^{V-1}, \qquad \text{(ii) } \mathcal{Z}_{\text{tok}} = \mathbb{R}^d \text{ (token-wise codebook)}, \qquad \text{(iii) } \mathcal{Z}_{\text{ctx}} = \mathbb{R}^d \text{ (contextualized)}.$$

For sequence length $L$, these apply positionwise with shared or tied encoders/decoders.

**Probability simplex.** $\mathcal{Z}_\Delta = \Delta^{V-1}$ models calibration-ready probabilistic targets, and the score functions may have clear information-geometry meanings (e.g., Fisher geometry on the simplex). However, the high-dimensionality and the valid distribution constraints make it hard to be directly generated (as the diffusion target). Furthermore, data are concentrated on simplex vertices (one-hots), resulting highly non-smooth denoising targets.

**Token-wise embedding with codebook.** Training targets for token-wise embedding space are codebook vectors $z_0 = e_{x_0} \in C \subset \mathbb{R}^d$ and diffusion evolves in $\mathbb{R}^d$. Decoding can be nearest-neighbor or softmax over codebook energies: $p(x \mid z) \propto \exp\{-\|z - e_x\|^2/\tau\}$ as in previous work (Li et al., 2022). When $d \ll V$, the lower dimensionality and the Euclidean geometry (naturally suitable for Gaussian noises) enable easier diffusion optimization. Nevertheless, the optimization target is still non-smooth due to the atomic codebook vectors and the naive decision boundaries. Most importantly, we now show that $\mathcal{Z}_{\text{tok}}$ is not more powerful than $\mathcal{Z}_\Delta$.

**Proposition E.1** (Representational power w.r.t. terminal classification)**.** *Any $z$-based classifier can be emulated by modeling logits in $\mathbb{R}^V$ and passing through a softmax with a learned final linear layer whose columns are $\{e_v\}$. Hence token-wise embeddings are not strictly more expressive than simplex-based logits for terminal token prediction.*

*Proof.* (Constructive.) Define logits $\ell(z) = W^\top z + b$ with columns of $W$ equal to code vectors (optionally re-centered); softmax$(\ell(z))$ approximates any energy-based decoder over the codebook. $\square$

The construction is straightforward since $\mathcal{E}_{tok} = \begin{bmatrix} e_1 \\ \dots \\ e_V \end{bmatrix} \in \mathbb{R}^{V \times d}$ is column full rank, thus any $z$ could be linearly reconstructed by a logit $\ell$ with the help of the codebook.

*Remark* E.2 (Why training can still be harder). The regression target $z_0 \in C$ makes the conditional expectation $\mathbb{E}[z_0 \mid z_t]$ a weighted average of code vectors, which lies *between* modes. Nearest-neighbor decoding then introduces a quantization gap; gradients around Voronoi boundaries are poorly conditioned, explaining larger prediction errors in practice.

**Contextualized embedding.** Here $z_0 = \mathcal{E}_{\text{ctx}}(x_0; \text{sequence context})$ depends on the whole sequence (and position). A decoder $D_{\text{ctx}}(z, \text{context}) \to \Delta^{V-1}$ maps the embedding back to token probabilities. If $\mathcal{E}_{\text{ctx}}$ is well-defined (which is true for state-of-the-art pretrained LLMs), the generation targets now contain rich information and themselves being smoother. Sequence-level information and long-range dependencies are encoded into a compact space, facilitating diffusion generation. The largest obstacle now becomes the decoding ambiguity, which could be tackled with the help of the discrete component.

*Remark* E.3 (Sufficiency and decoding). If $z_0 = \mathcal{E}_{\text{ctx}}(x_0, \text{ctx})$ is (approximately) a sufficient statistic for predicting $x_0$ given context (i.e., $p(x_0 \mid \text{ctx}, z_0) = p(x_0 \mid \text{ctx})$), then there exists a decoder $D_{\text{ctx}}$ achieving Bayes-optimal token prediction. In practice, approximate sufficiency improves sample efficiency but is not guaranteed, leading to residual ambiguity.

# F. Generated Examples

We attach some generated examples of CCDD-MoEDiT trained with Qwen3-Embedding-0.6B on OWT. The inference length and the number of denoising steps are both set to 512, with DDPM update for continuous tokens and MDLM posteriors for discrete tokens. The CFG guidance scale is 1.5, and the sampling temperature is 1.0 by default.

Her pancakes seemed to be far too bland, and it was clearly an amazing piece of cake. She falls for the pieces like this, as she starts to take a look at how she had grown up with I mean, this man's and the whole life he has taught her in her life. Laughing at her for not being representative of what she was never originally supposed to be given to that life, and that she was not really as miserable as what she had grown up, because no matter what people said, she did what she was supposed to be like to bring her onto her own world, the Land, but he keeps treating her and she goes one by one and somehow spins around and begins something, tries to play the extreme with the lives without generally treating her with what she was supposed to be like too. But at the end of the day, she is not having any enjoyment of that life and she seems to have the crap transition that should work for them. The poor state of their world between them and her being something somewhat different.

That actually not only hurts her, it just make its villains really stupid about her, and even worse than what they thought she was. That was simply a very normal combination of things, as she was incorporating a few devices into her life where she could still do anything, anything for a while, and she really was like she had to have a home. As I was lived along at that time, I started knowing that she had a really, really really shitty home, that they'd had nothing like interacting with me until her home was eventually taken over, and she was never physically or even physically interacting with me, and all this in those moments, I was really accepting that I was just doing what I was doing, and things did seem to don't help me as she'd always have, and she wasn't the family that she was supposed to be home to until the moment, only now did I know what I was doing, which was a really crappy home, and I was just given the task, unable to get any benefit from her. She didn't have such a role as as long as she finished, she had no idea what to do, she just had a real, shitty home. What she could not have had when was the current state of the home. My husband had been living with me to the point where his entire life was going to be so completely acceptable to his home that meant he was going to just move again, and just have all this fun in the home world.

Back to farms, there was a lot of different stuff about it, especially about after the farm had been changed and we had no idea what part of the culture and/or anything like it was, while the people that seemed to argue towards us went and in a subsequent meeting of my farm was worthless and I all realized that anyway it had been taken away since we had 90% of money in direct sales and we had no interaction with any other farmers or the manager which was what was actually happening for us. That's when I noticed a lot of the business around me and thought immediately, everything seemed very impersonal to the small restaurants it was on both iterations were that were both profitable and varied but this was open sourced for them to be innovative, I definitely would have seen something completely different than anyone else on earth and most likely not much of the possibility they were trying to have to it were really something that was available to them and not as innovative as we had ever dreamed of being, with such a limited options and the race around was how it looked on the farm and we can't even remember how it was brought in deeper than the bones with their fresh clothes but when anything turned a apart and it seemed to go further than how they went about the business, I begin to compare only myself to this interconnected world rather than make me get to a point where I would have reasonable concerns about anything like it. At first it seemed sort of seemed like a new outpost was any sign of plans on existing or more of it and not much later, much later it was so much that the front screen seemed to me very warped and just a sense of disgust his face was due to his colors nothing was really very apt to present for me to see what they were doing there all along and most of me that I had been getting into the community behind had anything I was capable of producing. It was within the last couple of months that I've seen a lot of poor people being able to rid themselves of this place, and all I keep mentioning is now since the small farms were seemingly only established in the food shops that time it's a question of my priorities and not addressing something about the animals is something I can argue with, but this years it's been one of my priorities and lately like my boss have been distracted with very rarely the moves that moved. I've once again seen another homeless restaurant that seemed to be basically within the framework, several times and in the last few years has changed for the community in Syracuse and the only one.

The world crop is evolving because it's such a very different process and if it is never going to be sustainable, that is something like that that is way too bad for the people. And that's because it is so obvious that there is a pretty major issue on the planet that that is linked to that and most of the problems are a lot of that is happening on the roads, it is not happening for the plant, it is not happening for the home and there is a whole lot of salt all over the Pacific oceans, and for some of it that is done to the oceans, and why are they just looking at the government, seeing all the roads and the lumber, and the sand beaches and stuff and all of those is happening every day and everybody knows they aren't going to do it but this is one of my feelings and I'm going to reduce it to that abstract and it is the face of the globe that they said this as much as they are, but the most important thing in that is that all of the existing plants, unlike the old progressive reactors, are not just at a 100% temperature like our corn and the stuff is just endlessly growing out, every single out one just because it is something that would be a problem so basically we will be just putting it that people can use as a source of their resources. Some of those crops can be cut off, we can have no irrigation, we can go back to farms, we can graft some of that things and it makes NO sense as they are all the same. See, if we have both crops as though they aren't the same then there is no reason that would be a problem for us, we know each other, but just because they are living different to each other and one thing that is a lot with those crops or that is some of these weeds can come out of soil and eventually it will be replaced with trees and so there will be a number of areas to which these plants would not have the kind of characteristics or truly some different degrees of proportion that they are having and others are nearly as effective that they are supposed to have. Of course, maybe that is where the Pre-Christian Christian Church came from but that is not something that is so obvious and it is extremely obvious but it is because the water in plants, that which belongs in the environment, so all the seeds taken from water are not there and the material is not formed on planets. Those parts are the good aspects of the process and are all we need, they can be enough.

