# OpenReview forum: "Coevolutionary Continuous Discrete Diffusion: Make Your Diffusion Language Model a Latent Reasoner"
_ICML.cc/2026/Conference — ICML 2026 regular_

### Official Review · Reviewer_92ua · 2026-02-27

**Soundness:** 2
**Presentation:** 3
**Significance:** 3
**Originality:** 3
**Overall Recommendation:** 4
**Confidence:** 4

**Summary:**

This paper presents a theoretical and empirical link between several of the most widely used paradigms for natural language modeling—namely diffusion models (both continuous and discrete) and LLMs. The authors argue that LLMs with latent reasoning (e.g., chain-of-thought) are strictly less expressive than Looped Transformers (LTs), which in turn are less expressive than Continuous Diffusion Models (CDMs). Discrete Diffusion Models (DDMs), a competing paradigm to LLMs, are likewise shown to be less expressive than CDMs. However, prior empirical results indicate that CDMs often underperform DDMs and LTs in practice. The authors attribute this discrepancy to trainability limitations, including the lack of intermediate supervision for LTs and optimization difficulties for CDMs. Motivated by these observations, the paper introduces Coevolutionary Continuous–Discrete Diffusion (CCDD), a diffusion framework that jointly models a continuous semantic space and a discrete token space via a coupled diffusion process. Experiments on natural language generation (LM1B and OWT) and reasoning benchmarks (including Sudoku, 3-SAT, and Countdown) are presented to demonstrate the benefits of combining the two modalities within a unified diffusion framework.

**Compliance With Llm Reviewing Policy:**

Affirmed.

**Final Justification:**

As in my original review, I think the paper's originality, significance and presentation are all good and has the potential of being accepted (after addressing my concerns). Now, since all concerns are addressed, I think this paper is worth a 4 (weak accept).

**Key Questions For Authors:**

1. More clarifications on the expressivity between CDMs and DDMs. Please correct me if I'm wrong. Definition 3.1 and Theorem 3.2 seem to place both CDMs and DDMs in a *continuous-space* framework, and then conclude that CDMs admit a larger trajectory family because they are supported over the full continuous domain, hence “more expressive” than DDMs. I’m not fully convinced this is a fair or task-aligned comparison. Since the downstream objective is to model *discrete* distributions (e.g., text), it would be more appropriate to analyze expressivity in the *discrete* domain. Moreover, as noted in [1], DDMs can already represent (in principle) any discrete distribution. In that sense, the claimed expressivity gap seem be overstated.
2. The following is quoted from Section 4.1. I’m not trying to be overly picky about language, but “preserve full semantics” sounds too absolute without direct experimental evidence or a precise operational definition. I would suggest toning this down.
> Thanks to the continuous component, CCDD is able to preserve full semantics in previous denoising steps
3. To see if CDM and DDM components do improve on each other, I think an ablation where either part is disabled is needed.

[1] Chen et al., 2024, Convergence Analysis of Discrete Diffusion Model: Exact Implementation through Uniformization

**Limitations:**

yes

**Strengths And Weaknesses:**

The paper is clearly written and well motivated.

The connections drawn among discrete diffusion models, continuous diffusion models, looped transformers and LLMs with latent reasoning are interesting and insightful and hence significance and originality are both good.

The paper is also technically sound in that it provides formal definitions, theoretical formulations, and proofs characterizing the expressivity of each model class. That said, some explanations are somewhat handwavy. I elaborate on these concerns in the Questions section below.

---

> ### Author Rebuttal · Authors · 2026-03-31
>
> Thank you for your feedback and suggestions. Here we would like to answer your questions as follows.
>
> ### **Q1: Expressivity comparison**
>
> We thank the reviewer for this important comment. We would like to clarify that our comparison is **not** made in the infinite-capacity regime where discrete diffusion models (DDMs) can be universal in principle, but rather under the setting adopted throughout the paper: **the same architecture and the same finite parameter budget**. Under this practically relevant constraint, the key issue is not only whether the model can represent an arbitrary discrete distribution in the limit, but whether the *iterative denoising computation* can preserve and manipulate sufficient information to realize rich *joint terminal distributions*.
>
> More specifically, even when the final target distribution is discrete, the reverse process matters. In a standard DDM, at each reverse step the model predicts logits, but the next state is obtained by *sampling discrete tokens* and feeding only those sampled tokens (finitely supported) into the next denoising step. This has two consequences. First, the next-step model does not receive the full logit vector, but only a sampled symbol, so as formalized by **Lemma~B.9**, the map from logits to sampled token is intrinsically information-losing, repeatedly applying an unavoidable hard bottleneck on the intermediate representation along the sampling trajectory. Second, the sampling is conducted token-wise (marginally), thus unmasking multiple tokens in a single step fails to model the joint distribution.
>
> By contrast, a CDM propagates the entire continuous latent state throughout the reverse process. Since no mandatory token sampling is inserted between denoising steps, the model can preserve fine-grained uncertainty and high-order cross-token dependence in the latent state and continue refining them jointly across time, directly modeling a **joint distribution over the entire latent state trajectory**, whereas the DDM reverse process is forced to factor its computation through sampled discrete states at every step. Therefore, although both models may eventually decode to discrete outputs, the CDM has access to a strictly richer class of intermediate computations and trajectory laws.
>
> The above theoretical claim is also supported by experimental observations. In **Table 4** of this [link for updated experimental results](https://anonymous.4open.science/r/CCDD-ICML-rebuttal-CB89/), CCDD with only 16 diffusion steps outperforms discrete MDLM baseline with 256 diffusion steps when generating sequences of length 512. This validates the strong capability to model joint distribution of continuous diffusion, and the superior practical expressivity of CCDD.
>
> To summarize, our claim is not that DDMs are non-universal in the abstract infinite-capacity sense, but that, in the finite-capacity regime considered in this paper, continuous diffusion has strictly stronger effective expressivity than discrete diffusion due to (i) the larger representation space, (ii) the absence of compulsory marginal token sampling inside the reverse loop, and (iii) the resulting ability to preserve and model joint information across the entire denoising trajectory.
>
> ### **Q2: Semantics in CDM**
>
> Thank you for pointing out - we are happy to adjust the expression. Our original intention is to describe the fact that *the input of CDM's next denoising step can directly be the output of the previous step*, which is impossible for DDM with inevitable information loss. As noted above, this observation is supported by both theory (**Lemma~B.9**) and experiments (Table 4 in https://anonymous.4open.science/r/CCDD-ICML-rebuttal-CB89/ shows that CCDD has significantly better few-step sample quality, thanks to the capability of preserving rich semantics and modeling the joint distribution).
>
> ### **Q3: Ablations of components**
>
> Actually, our original experiments already partially reflect this ablation. For example, CCDD-MDiT uses the same DiT architecture as MDLM discrete baseline, where the latter can be viewed as the case where continuous part is disabled. We additional train a purely continuous diffusion without discrete component. All models with same parameters are trained on LM1B for 500k iterations. This result shows that continuous component alone is hard to train, yet CCDD combines both components and yields substantial improvement of both modalities.
>
> | Model | Discrete | Continuous | CCDD |
> |-----|-----|-----|-----|
> | Valid. PPL | 40.87   | 164.19   | 30.26 |
>
> In addition, the CFG results in **Table 4** of our original submission already shows that at inference time, one can flexibly balance the discrete and continuous components. $w=0$ is the discrete-only setting, $w=1$ is the standard CCDD setting using both modalities, while $w>1$ corresponds to the cases where the model relies more on the continuous component. All CFG weights consistently outperforms MDLM baseline.

---

> > ### Author Rebuttal · Reviewer_92ua · 2026-04-02
> >
> > My concerns are addressed and I am adjusting my score to 4.

---

> > > ### Author Response · Authors · 2026-04-03
> > >
> > > Thank you for acknowledging our rebuttal - we are glad your concerns were addressed. We appreciate your feedback and will include these new results in our revision.

---

### Official Review · Reviewer_1y54 · 2026-03-05

**Soundness:** 4
**Presentation:** 3
**Significance:** 3
**Originality:** 2
**Overall Recommendation:** 4
**Confidence:** 4

**Summary:**

This paper proposes coevolutionary continuous discrete diffusion (CCDD), a framework that combines continuous diffusion and discrete diffusion. In diffusion-based language modeling, the widely adopted setting is typically discrete diffusion, while continuous diffusion has generally shown weaker performance. In contrast, this paper revisits and argues (with supporting analysis/proofs) that continuous diffusion can offer stronger expressivity than discrete diffusion, including the ability to simulate looped transformers, while identifying training difficulty as its main practical bottleneck. To address this, the paper introduces a hybrid framework that mixes discrete and continuous diffusion, aiming to simultaneously inherit the theoretical expressivity of continuous diffusion and the practical trainability of discrete diffusion.

**Compliance With Llm Reviewing Policy:**

Affirmed.

**Final Justification:**

The theory is strong, the paper provides valuable insights across a broad range of architectures, and the additional experiments in the rebuttal further reinforce its contributions.

**Key Questions For Authors:**

Please carefully check and address the weaknesses raised above. Additionally, I have a question to authors: was there no way to train the encoder as well? Since the encoder must be pre-trained, this inevitably makes the discrete diffusion rely on an autoregressive model, and hinders discrete diffusion from being a stand-alone framework. Furthermore, it may introduce dataset bias, and it is also unclear whether keeping the encoder fixed is optimal.

**Limitations:**

The impact statements section is written; yet I suggest discussing W2 and W3 above.

**Strengths And Weaknesses:**

## **Strengths**
Basically, I found Section 3 to be a major contribution, and I appreciate the authors for providing such intuitions with theorems.
1. Theorem 1 provides a novel and insightful proof that continuous diffusion is more expressive than discrete diffusion, and the accompanying intuitive explanation for readers (logits quantization) is convincing. It is also interesting that Proposition 3.4 shows that a continuous diffusion model can simulate a looped transformer.
2. Table 1, together with the detailed discussion, provides a clear and convincing explanation of why a wide variety of continuous diffusion is fundamentally harder to train.
3. The forward/reverse processes of CCDD are simple, and the formulation itself appears generalizable and scalable.

## **Weaknesses**
While the theory and the overall framework are insightful, the main issues arise in the practical realization and empirical results starting from Section 4.2.
1. The three model architectures proposed to implement CCDD are not compelling.
* In MDiT, the parameters in the DiT backbone are fully shared between discrete token prediction and continuous embedding prediction. Intuitively, it is unclear whether these two signals can be optimally learned within the same parameter space. In contrast, MMDiT/MoEDiT separate them, but this inevitably increases the parameter count.
* These drawbacks are reflected in the experimental results. While the method performs well in the LM1B setting (Table 1), where the training dataset is smaller and the model length is shorter, its performance on OWT does not appear strong. Even with substantially more parameters, CCDD-MoEDiT / CCDD-MMDiT do not show a clear improvement in PPL over MDLM. Moreover, Gen NLL, which better reflects actual text generation quality, is only reported for CCDD-MoEDiT. In the most theoretically motivated setting (w=0), the gain over MDLM is marginal despite using more parameters.
2. The experimental evidence is partial, making it difficult to assess whether the method is effective.
* The three variants (MDiT, MoEDiT, MMDiT) are evaluated inconsistently across settings: MDiT has no OWT results, MoEDiT has no LM1B results, and MDiT/MMDiT does not report Gen NLL. As noted in W1, it is not obvious that these architectures should work well, and the incomplete evaluation exacerbates this concern.
* Key metrics in diffusion language modeling are missing: there is no few-step Gen NLL and no zero-shot PPL. In diffusion language modeling [1, 2, 3], it is standard to report both. Few-step Gen NLL is particularly important because a core motivation of diffusion LMs is to move beyond one-by-one token generation in autoregressive models. Zero-shot PPL is also widely used as a sanity check against dataset-specific overfitting. In fact, I have often seen papers [1, 3] report only zero-shot PPL, but rarely papers that report only test PPL without zero-shot PPL. This is especially important here because the approach relies heavily on a pre-trained encoder, whose effectiveness may vary substantially across datasets; zero-shot PPL would help clarify this.
* The parameter sizes across the compared models differ too much. In particular, CCDD-MMDiT uses more than twice as many parameters as MDLM, and it seems plausible that scaling MDLM to a comparable parameter budget could outperform CCDD.

[1] Sahoo et. al., "Simple and Effective Masked Diffusion Language Models," NeurIPS'24.

[2] Sahoo et. al., "The Diffusion Duality," ICML'25.

[3] Xie et. al., "Variational Autoencoding Discrete Diffusion with Enhanced Dimensional Correlations Modeling," ICLR'26.

3. There is no comparison to other related work [4, 5] that combines continuous and discrete diffusion models.
* I acknowledge that those papers may be concurrent work, and I am not trying to diminish the novelty of this paper. However, the key distinction appears to be whether one decouples the discrete and continuous spaces or models them jointly. This should be explained more clearly to readers, along with the respective pros/cons of each design choice.
* Compared to those concurrent works, the empirical performance of this paper looks weaker. Again, I do not want to disparage CCDD, given that the directions may be orthogonal, and the works are concurrent. Still, the paper should explain what theoretical/algorithmic value or future impact this method provides relative to those works, even if it is not the strongest empirically at present.

[4] Pynadath et. al., "CANDI: Hybrid Discrete-Continuous Diffusion Models," Arxiv'25.

[5] Zheng et. al., "Continuously Augmented Discrete Diffusion model for Categorical Generative Modeling," ICLR'26.

Overall, I think the paper is well written in the sense that the problem is clearly defined, the theoretical motivation is strong, and the proposed framework is intuitive and simple. However, the actual model designs and the resulting empirical evidence are not convincing, so I currently lean toward weak rejection. If the authors fill in the missing experiments in W2 and provide a more thorough discussion of the practical trade-offs, I would be willing to raise my score. Moreover, if additional experiments show competitive or superior performance relative to existing approaches, I would be open to an even higher score.

---

> ### Author Rebuttal · Authors · 2026-03-31
>
> We sincerely thank the reviewer for the comments and insights. We address your concerns as follows.
>
> ### **W1: Architectures**
>
> Thanks for the note. Due to limited space of the main text, we actually provide detailed comparison of architectures in Appendix C.2 of our original submission. **Figure 4** already demonstrates the computation FLOPs and parameter counts of different architectures. To summarize, CCDD with MDiT is able to significantly surpass baselines without increasing non-embedding parameters and FLOPs; despite more parameters and FLOPs, MMDiT and MoeDiT both showcase much stronger performance compared with discrete baselines. Therefore, MMDiT balances efficiency and performance, and MoEDiT balances parameters and performance.
>
> We want to clarify that we report the performance on OWT in Table 3 for two different tokenizers. When both trained with Qwen2 tokenizer, CCDD-MDiT reduces 20\% ppl compared with MDLM baseline with almost no additional FLOPs or parameters; other architectures further improves performance by scaling FLOPs or parameters.
>
> For Gen NLL, $w=0$ is not the most theoretically motivated setting - it is the *discrete only inference mode*, which shows that continuous training alone (even discarded during inference) could be helpful for the discrete part. The *standard CCDD* actually corresponds to $w=1$, which shows significant improvement over MDLM; with $w=1.5$, CCDD provides more powerful results, proving the flexibility of inference and the *effectiveness of CFG*.
>
> ### **W2: Experiments**
>
> During rebuttal, we add extensive new experiments with significant improvements to address every piece of your concern; the results are available at:
>
> https://anonymous.4open.science/r/CCDD-ICML-rebuttal-CB89/
>
> - Comprehensive evaluations of architectures on both datasets: Due to limited time, we train all models for 500k iterations, corresponding to approximately 17B tokens for LM1B and 65B tokens for OWT (half of the original pretraining tokens). As shown in **Table 1** and **Table 2** in the updated pdf, all architectures consistently outperforms MDLM baseline. Notably, we adjust the model size too keep parameter counts in a comparable range.
>
> - Few-step Gen. NLL: Indeed, this is an important metric. We evaluate few-step generative NLL, and present the results in **Table 4** in the link. The sequence length is set to 512, and CCDD with only 16 diffusion steps shows lower Gen. NLL compared to MDLM with even 256 steps. This $16\times$ speed up effectively shows the few-step generation quality of CCDD, thanks to the joint distribution modeling capacity and superior expressivity of the continuous component.
>
> - Zero-shot PPL: Actually, we already report zero-shot PPL of downstream benchmarks in **Table 8 of our original submission**. We additionally provide a comprehensive version in **Table 3 of the updated link**: both CCDD variants outperform discrete MDLM/GIDD (even with larger size) baselines on almost all datasets, even outperforming AR models in some benchmarks.
>
> - Justification of parameters: In our additional experiments (Table 1 and Table 2 in the updated link), we adjust the model size too keep parameter counts in a comparable range. With *ignorable or moderate more parameters*, CCDD with all architectures **consistently outperforms MDLM baseline with a large margin**.
>
> ### **W3: Concurrent work**
>
> Thank you for the note, we will definitely discuss these concurrent papers and analyze the difference in the revised version. CCDD leverages a joint modeling of both modalities, which is a principled an expressive way to incorporate the advantages of continuous diffusion. We believe the current version of CCDD is not only more theoretically grounded, but also empirically valid/comparable to these papers.
>
> ### **Q1: Co-training encoder**
>
> This is an interesting question. Standard (latent) diffusion models always build on pretrained, high-quality latent space, for example, the pretrained SD-VAE for image diffusion. It is unusual to use a random latent space due to the potential problems in generated semantics and reconstruction/decoding quality. However, with recent techniques such as REPA-E[6] or UNITE[7], we believe it is possible to co-train the encoder, though the experiments are out of the scope of this paper.
>
> We would also like to emphasize that **being able to leverage pretrained latent space's knowledge is exactly the advantage of our methods**. As explained in **Section 4.2**, we are the first to utilize representation guidance strategy in the context of discrete diffusion and text generation, and approach significantly accelerates training convergence.
>
> [6] Leng, Xingjian, et al. "Repa-e: Unlocking vae for end-to-end tuning of latent diffusion transformers." Proceedings of the IEEE/CVF International Conference on Computer Vision. 2025.
>
> [7] Duggal, Shivam, et al. "End-to-End Training for Unified Tokenization and Latent Denoising." arXiv preprint arXiv:2603.22283 (2026).

---

> > ### Author Rebuttal · Reviewer_1y54 · 2026-04-03
> >
> > Thank you for the detailed rebuttal. The additional experiments have addressed my concerns. I will raise my score to 4.
> >
> > To provide broader insight to the community, I would appreciate it if, in the event of acceptance, the authors could extend the experiments given in rebuttal to 1M steps and include those results in the main paper.

---

> > > ### Author Response · Authors · 2026-04-03
> > >
> > > We sincerely thank the reviewer for the feedback and glad that we have addressed your concerns. We will include these new results, as well as pretraining up to 1M steps in the revised version.

---

### Official Review · Reviewer_2PKA · 2026-03-11

**Soundness:** 3
**Presentation:** 2
**Significance:** 3
**Originality:** 2
**Overall Recommendation:** 4
**Confidence:** 4

**Summary:**

The authors first compare continuous diffusion, discrete diffusion, and looped Transformers from a theoretical perspective, arguing that continuous diffusion has stronger expressive power at the trajectory level. Then, they analyze why continuous diffusion often underperforms discrete diffusion in practice despite its theoretical advantages. Based on this analysis, they propose CCDD, which constructs a joint diffusion process over both the discrete token space and the continuous representation space, and performs denoising on both modalities simultaneously. Through joint denoising, pretrained contextual embeddings, and related designs, the method attempts to balance expressiveness and trainability.

**Compliance With Llm Reviewing Policy:**

Affirmed.

**Final Justification:**

Thank you to the authors for their thoughtful revisions and clarifications, which have improved the overall quality and presentation of the paper. The work addresses an interesting problem and demonstrates solid technical effort.

**Key Questions For Authors:**

1. Please revise the spelling and wording errors throughout the manuscript, add necessary explanations, and unify the names of the baselines, or explain why different baselines are used in different validation experiments.
2. Please strengthen the experimental analysis, for example by clarifying the main sources of the performance gains.
3. Tables 3 and 8 show that GPT-2 and LLaMA outperform CCDD in multiple scenarios. Could the authors explain the reasons for this?

**Limitations:**

Yes

**Strengths And Weaknesses:**

Strength:
1. The manuscript provides a complete line of argument, combining theoretical comparison and proof with empirical ablations on representation-level choices.
2. The overall structure of the manuscript is clear. From problem formulation to theoretical analysis, method design, and experimental validation, the presentation is generally coherent, and the main algorithm is overall understandable and reproducible.
3. The manuscript discusses how to better balance the expressiveness of continuous latent spaces and the trainability of discrete generation in diffusion language models, which is of value.

Weakness:
1. The manuscript does not clearly verify how much of the improvement of CCDD comes from the joint diffusion mechanism itself and how much comes from the pretrained representations. If the main gains are primarily due to pretrained representations, then the comparability with other methods that do not use pretrained representations becomes questionable.
2. The proposed framework includes the term “Coevolutionary” in its name, but the framework does not appear to involve a multi-model setting, nor does the manuscript clearly explain where and in what sense coevolution is actually used.
3. The manuscript contains spelling and wording errors such as “semnatics” and “sumarized.” In addition, the presentation of results is confusing, lacks visualization, and uses inconsistent baseline names across different experiments. These issues significantly reduce the readability of the paper. If the authors intentionally used different baselines in different experiments, a clear justification is necessary.
4. Although the problem addressed in the manuscript is itself important, the current empirical validation is still mainly limited to medium-scale pretraining and relatively controlled reasoning tasks. The paper does not yet provide sufficient evidence regarding performance in larger-scale language modeling, open-ended generation, long-context settings, or scenarios closer to real-world applications.
5. Continuous diffusion models and contextual embeddings from pretrained large models are both existing techniques. The authors do not seem to introduce a fundamentally new paradigm; rather, the method appears more like a combination of existing techniques.

---

> ### Author Rebuttal · Authors · 2026-03-31
>
> Thank you for your feedback and suggestions. We would like to address your concerns as follows.
>
> ### **W1 \& W5 \& Q2: Pretrained representations and paradigm**
>
> The improvement comes from both representation guidance and the expressive joint diffusion framework. As emphasized in Section 4.2 of the paper, we believe that leveraging pretrained representation itself is a novel technique that is valuable -- **being able to leverage pretrained latent space's knowledge** is exactly the advantage of our methods. Similar work includes RCG[1], REPA[2], REED[3] etc. All these papers leverage pretrained representations to improve the quality of generative models. Our paper shares similar spirit, yet **we are the first one to propose similar representation guidance paradigm in the context of language generation**, specifically with diffusion language models. We also provide rigorous theory for the joint training procedure, see the entire **Section B.2**.
>
> Experimentally, leveraging high-quality pretrained embeddings significantly accelerates diffusion model training: on the LM1B dataset, it takes MDLM 1000k iterations to achieve the $39.17$ perplexity, while CCDD needs only 40k iterations, resulting in a **$25\times$ acceleration**. Also as presented in our reply to Reviewer 92ua, combining discrete and continuous diffusion could lead to *much better performance than both*. All these facts are significant enough to support that this paradigm is not a naive combination of existing techniques, but a **fundamentally novel and principled framework** that **improves expressiveness and accelerates training convergence**. We firmly believe that both our theoretical and empirical results are non-trivial.
>
> Moreover, standard (latent) diffusion models always build on pretrained, high-quality latent space, for example, the pretrained SD-VAE for image diffusion. However, with recent techniques such as REPA-E[4], we believe it is possible to co-train the encoder, though the experiments are out of the scope of this paper.
>
> [1] Tianhong Li, et al. Return of unconditional generation: A self-supervised representation generation method. NeurIPS 2024.
>
> [2] Sihyun Yu, et al. Representation alignment for generation: Training diffusion transformers is easier than you think. ICLR 2025.
>
> [3] Chenyu Wang, et al. Learning diffusion models with flexible representation guidance. NeurIPS 2025.
>
> [4] Leng, Xingjian, et al. "Repa-e: Unlocking vae for end-to-end tuning of latent diffusion transformers." CVPR 2025.
>
> ### **W2: Definition of coevolutionary**
>
> Actually, the term "coevolutionary" is appearing in the sense of the continuous and discrete part evolve together in the diffusion process. As already explained in the paper, the factorized parameterization is sufficiently expressive to model the joint distribution in the union of discrete and continuous space. We believe the joint diffusion is a principled and powerful framework to incorporate the advantages of continuous diffusion into discrete modality.
>
> ### **W3 \& Q1: Typos**
>
> We will correct the typos and try to unify the names. Can you point out in detail why the presentation of results is confusing? Actually, most names of the baselines are fully consistent (e.g., MDLM, GIDD etc.), except from Table 5 since it is a completely different dataset and pretraining task, while baseline names are taken from previous work of Ye etal., 2024a.
>
> ### **W4 \& Q2: Experiments**
>
> During rebuttal, we add extensive new experiments to showcase the effectiveness of CCDD across many tasks, datasets and metrics, including pretraining, downstream benchmarks and open-ended generation. Results are available at this link:
>
> https://anonymous.4open.science/r/CCDD-ICML-rebuttal-CB89/
>
> As many other academic papers, it is impossible to include all these topics -  each of them are worth a separate publication and is beyond our scope. However, we believe these results of pretraining and zero-shot downstream question answering are already close enough to real-world scenarios and sufficient to validate the superiority of CCDD.
>
> ### **Q3: Comparison with AR**
>
> Our goal is not to surpass AR in all tasks, which is not achievable by any diffusion language models. Our main goal is to improve discrete diffusion with continuous component, and every single experiment supports that **CCDD consistently outperforms discrete baselines such as MDLM**. Moreover, CCDD already has very close performance to AR models, and excels in complex reasoning tasks such as Sudoku, SAT and Countdown etc (Table 5 in our submission).
>
> Hope that our clarification has answered your questions, and we are happy to participate in any follow-up discussions.

---

> > ### Author Rebuttal · Reviewer_2PKA · 2026-04-04
> >
> > Thanks for clarifying. I will maintain my score.

---

> > > ### Author Response · Authors · 2026-04-05
> > >
> > > Dear reviewer,
> > >
> > > We appreciate your time and your review. While we understand that you have a large workload, we would like to politely point out that your rebuttal acknowledgement is not self-consistent, and may be a violation to the ICML reviewer code of conduct. In particular, you choose (b) partially resolved - have follow-up questions, however in the justification you do not explicitly state any follow-up questions, yet choose to maintain the score without any reasons. We believe your justification is insufficient which makes your rating less convincing.
> > >
> > > We worked hard during rebuttal and we believe that our rebuttal has addressed all the previously raised concerns. For training paradigms and the use of pretrained representations, we clarify that CCDD is a **novel, rigorous and universal training paradigm** that combines discrete and continuous modalities; moreover, being able to utilize high-quality pretrained representations **significantly accelerates diffusion training**. For experiments, we provide **extensive new experiments with state-of-the-art results** spanning across pretraining validation perplexity, few-step generative ppl, and downstream evaluations on a wide variety of tasks and benchmarks, which are available at this link https://anonymous.4open.science/r/CCDD-ICML-rebuttal-CB89/.
> > >
> > > Notably, all other reviewers have explicitly acknowledged that our rebuttal have fully addressed their concerns, and raised their score all to positive. We kindly inquire if you have any follow-up questions which we will be happy to answer, or acknowledge that our rebuttal has addressed your concerns and adjust the score accordingly. Thank you very much!
> > >
> > > Sincerely,
> > >
> > > Authors

---

### Official Review · Reviewer_E2PW · 2026-03-13

**Soundness:** 4
**Presentation:** 4
**Significance:** 3
**Originality:** 3
**Overall Recommendation:** 6
**Confidence:** 4

**Summary:**

In this paper, the authors present a unified approach for continuous and discrete diffusion on discrete token spaces. Theoretically, the authors demonstrate that discrete diffusion after embeddings is strictly less expressive than continuous diffusion, which can express looped transformers. However, training continuous language diffusion is more difficult than discrete, so the authors propose uniting the two by noising the discrete tokens both in a discrete manner as well as continuously through the embedding space. Performance on complex toy reasoning tasks see very large boosts in performance.

**Compliance With Llm Reviewing Policy:**

Affirmed.

**Final Justification:**

The rebuttal addressed all my concerns and I maintain my original positive score.

**Key Questions For Authors:**

- For the results for diffusion language models in Table 5, are the results achieved through random order unmasking of the discrete tokens, or is the positional order of unmasking chosen greedily (according to some uncertainty metric)?

**Limitations:**

yes

**Strengths And Weaknesses:**

Strengths:

- The idea of combining discrete and continuous diffusion is very natural and strongly motivated by observations of representation strength vs. trainability.

- Performance boosts on both toy reasoning tasks as well as general language benchmarks are strong relative to baseline approaches.

- Strong theoretical support and motivations for effective empirical results.

Weaknesses:

- While the authors do exhibit boosted performance on general language understanding and question-answering capabilities, an evaluation on more standard reasoning benchmarks like MATH500/AIME would be helpful to demonstrate whether the stronger reasoning from toy tasks like Sudoku transfer over to math.

---

> ### Author Rebuttal · Authors · 2026-03-31
>
> We sincerely thank the reviewer for the positive review. Here we would like to answer your questions as follows.
>
> **Reasoning benchmarks**: our experimental settings follow previous academic-scale pretraining work, including but not limited to SEDD, MDLM, GIDD etc. These reasoning benchmarks would require industrial-scale pretraining, including much larger models, tens of thousands of GPUs, extensive high quality private data and months of training -- all of which are beyond our scope. We believe the current experiments are sufficient to validate our theory and showcase the superiority of our methods, and scaling CCDD alone and specifically adapt to mathematical reasoning alone worth another independent publication. We also provide additional experimental results during rebuttal: https://anonymous.4open.science/r/CCDD-ICML-rebuttal-CB89/
>
> **Decoding order**: in Table 5, we use confidence-based decoding order, i.e., a positional unmasking order chosen greedily according to token entropy. This strategy completely follows Ye etal., 2024a.
>
> Ye, J., Gao, J., Gong, S., Zheng, L., Jiang, X., Li, Z., and Kong, L. Beyond autoregression: Discrete diffusion for complex reasoning and planning. arXiv preprint, arXiv:2410.14157,2024a.
>
> We are happy to answer any follow-up questions.

---

> > ### Author Rebuttal · Reviewer_E2PW · 2026-04-02
> >
> > I thank the authors for their rebuttal. My questions have been addressed, and I maintain my original positive score.

---

> > > ### Author Response · Authors · 2026-04-02
> > >
> > > We sincerely appreciate the reviewer's positive review and instant reply. Thank you!

---

### Decision · Program_Chairs · 2026-04-30

**Decision:**

Accept (regular)

**Comment:**

This paper proposes CCDD, a framework that jointly models continuous and discrete diffusion for language generation. The theoretical contribution — showing that continuous diffusion is strictly more expressive than discrete diffusion at the trajectory level while identifying trainability as the practical bottleneck — is recognized by all reviewers as strong and insightful. The proposed solution of coupling both modalities in a joint diffusion process is well-motivated and technically sound. All four reviewers gave positive assessments after the rebuttal. I therefore recommend acceptance.